# Synthetic Data Generators – Sequential and Private

**Olivier Bousquet**
Google Research, Brain team
Zürich
obousquet@google.com

**Roi Livni**
Tel-Aviv University
Tel-Aviv, Israel
rlivni@tauex.tau.ac.il

**Shay Moran**
Technion
Haifa, Israel
shaymoran1@gmail.com

## Abstract

We study the sample complexity of private synthetic data generation over an unbounded sized class of statistical queries, and show that any class that is privately proper PAC learnable admits a private synthetic data generator (perhaps non-efficient). Previous work on synthetic data generators focused on the case that the query class $\mathcal{D}$ is finite and obtained sample complexity bounds that scale logarithmically with the size $|\mathcal{D}|$. Here we construct a private synthetic data generator whose sample complexity is independent of the domain size, and we replace finiteness with the assumption that $\mathcal{D}$ is privately PAC learnable (a formally weaker task, hence we obtain equivalence between the two tasks).

## 1 Introduction

Generating differentially–private synthetic data [9, 14] is a fundamental task in learning that has won considerable attention in the last few years [22, 34, 23, 16].

Formally, given a class $\mathcal{D}$ of distinguishing functions, a fooling algorithm receives as input IID samples from an unknown real-life distribution, $p_{real}$, and outputs a distribution $p_{syn}$ that is $\epsilon$-close to $p_{real}$ w.r.t the *Integral Probability Metric* ([29]), denoted $\mathrm{IPM}_{\mathcal{D}}$:

$$\mathrm{IPM}_{\mathcal{D}}(p, q) = \sup_{d \in \mathcal{D}} \left| \mathop{\mathbb{E}}_{x \sim p} [d(x)] - \mathop{\mathbb{E}}_{x \sim q} [d(x)] \right| \tag{1}$$

A DP-SDG is then simply defined to be a differentially private fooling algorithm.

A fundamental question is then: Which classes $\mathcal{D}$ can be privately fooled? In this paper, we focus on sample complexity bounds and give a first such characterization. We prove that a class $\mathcal{D}$ is DP–foolable if and only if it is privately (proper) PAC learnable. As a corollary, we obtain equivalence between several important tasks within private learning such as proper PAC Learning [25], Data Release [14], Sanitization [6] and what we will term here *Private Uniform Convergence*.

Much focus has been given to the task of synthetic data generation. Also, several papers [5, 23, 16, 20, 21] discuss the reduction of private fooling to private PAC learning. In contrast with previous work, we assume an arbitrary large domain. In detail, previous existing bounds normally scale logarithmically with the size of the query class $\mathcal{D}$ (or alternatively, depend on the size of the domain). Here we initiate a study of the sample complexity that does not assume that the size of the domain is fixed. Instead, we only assume that the class is privately PAC learnable, and obtain sample complexity

bounds that are independent of the cardinality $|\mathcal{D}|$. We note that the existence of a private synthetic data generator entails private proper PAC learning, hence our assumption is a necessary condition for the existence of a DP-SDG.

The general approach taken for generating synthetic data (which we also follow here) is to exploit an online setup of a sequential game between a generator that aims to fool a discriminator and a discriminator that attempts to distinguish between real and fake data. The utility and generality of this technical method, in the context of privacy, has been observed in several previous works [22, 32, 20]. However, in the finite case, specific on-line algorithms, such as *Multiplicative Weights* [20] and *Follow-the-Perturbed-Leader* [37] are considered. The algorithms are then exploited, in a white-box fashion, that allow easy construction of SDGs. The technical challenge we face in this work is to generalize the above technique in order to allow the use of no-regret algorithms that work over infinite classes. Such algorithms don't necessarily share the attractive traits of MW and FtPL that allow their exploitation for generating synthetic data. To overcome this, we study here a general framework of *sequential SDGs* and show how an *arbitrary* online algorithm can be turned, via a Black-box process, into an SDG which in turn can be privatized. We discuss these challenges in more detail in the full version [10].

Thus, the technical workhorse behind our proof is a learning primitive which is of interest of its own right. We term it here *Sequential Synthetic Data Generator* (Sequential-SDG). Similar frameworks appeared [20, 37] in the context of private-SDGs but also more broadly in the context of generative learning [19, 27, 18, 17]. We further discuss this deep and important connection between private learning and generative learning in Section 5

In the sequential-SDG setting, we consider a sequential game between a generator (player G) and a discriminator (player D). At every iteration, player G proposes a distribution and player D outputs a discriminating function from a prespecified binary class $\mathcal{D}$. The game stops when player G proposes a distribution that is close in $\text{IPM}_{\mathcal{D}}$ distance to the true target distribution. As we focus on the statistical limits of the model, we ignore the optimization and computational complexity aspects and we assume that both players are omnipotent in terms of their computational power.

We provide here characterization of the classes that can be *sequentially fooled* (i.e. classes $\mathcal{D}$ for which we can construct a sequential SDG) and show that the sequentially foolable classes are exactly *Littlestone classes* [28, 7]. In turn, we harness sequential SDGs to generate synthetic data together with a private discriminator in order to generate private synthetic data. Because this framework assumes only a private learner, we in some sense show that the sequential setting is a canonical method to generate synthetic data.

To summarize this work contains several contributions: We provide the first domain-size independent sample complexity bounds for DP-Fooling, and show an equivalence between private synthetic data generation and private learning. Second, we introduce and characterize a new class of SDGs and demonstrate their utility in the construction of private synthetic data.

## 2   Primineries

In this section we recall standard definitions and notions in differential privacy and learning (a more extensive background is also given in the full version [10]). Throughout the paper we will study classes $\mathcal{D}$ of boolean functions defined on a domain $\mathcal{X}$. However, we will often use a dual point of view where we think of $\mathcal{X}$ as the class of functions and on $\mathcal{D}$ as the domain. Therefore, in order to avoid confusion, in this section we let $\mathcal{W}$ denote the domain and $\mathcal{H} \subseteq \{0,1\}^{\mathcal{W}}$ to denote the functions class.

### 2.1   Differential Privacy and Private Learning

Differential Privacy [13, 12] is a statistical formalism which aims at capturing algorithmic privacy. It concerns with problems whose input contains databases with private records and it enables to design algorithms that are formally guaranteed to protect the private information. For more background see the surveys [15, 35].

The formal definition is as follows: let $\mathcal{W}^m$ denote the input space. An input instance $\Omega \in \mathcal{W}^m$ is called a *database*, and two databases $\Omega', \Omega'' \in \mathcal{W}^m$ are called neighbours if there exists a single

$i \leq m$ such that $\Omega'_i \neq \Omega''_i$. Let $\alpha, \beta > 0$ be the privacy parameters, a randomized algorithm $M : \mathcal{W}^m \to \Sigma$ is called $(\alpha, \beta)$-differentially private if for every two neighbouring $\Omega', \Omega'' \in \mathcal{W}^m$ and for every event $E \subseteq \Sigma$:

$$\Pr\big[M(\Omega') \in E\big] \leq e^\alpha \Pr\big[M(\Omega'') \in E\big] + \beta.$$

An algorithm $M : \cup_{m=1}^{\infty} \mathcal{W}^m \to Y$ is called differentially private if for every $m$ its restriction to $\mathcal{W}^m$ is $(\alpha(m), \beta(m))$-differentially private, where $\alpha(m) = O(1)$ and $\beta(m)$ is negligible[1]. Concretely, we will think of $\alpha(m)$ as a small constant (say, 0.1) and $\beta(m) = O(m^{-\log m})$.

**Private Learning.** We next overview the notion of Differentially private learning algorithms [25]. In this context the input database is the training set of the algorithm.

Given a hypothesis class $\mathcal{H}$ over a domain $W$, we say that $\mathcal{H} \subseteq \{0, 1\}^{\mathcal{W}}$ is privately PAC learnable if it can be learned by a differentially private algorithm. That is, if there is a differentially private algorithm $M$ and a sample complexity bound $m(\epsilon, \delta) = \text{poly}(1/\epsilon, 1/\delta)$ such that for every $\epsilon, \delta > 0$ and every distribution $\mathbb{P}$ over $\mathcal{W} \times \{0, 1\}$, if $M$ receives an independent sample $S \sim \mathbb{P}^m$ then it outputs an hypothesis $h_S$ such that with probability at least $1 - \delta$:

$$L_{\mathbb{P}}(h_S) \leq \min_{h \in \mathcal{H}} L_{\mathbb{P}}(h) + \epsilon,$$

where $L_{\mathbb{P}}(h) = \mathbb{E}_{(w,y) \sim \mathbb{P}}\big[\mathbb{1}[h(w) \neq y]\big]$. If $M$ is *proper*, namely $h_S \in \mathcal{H}$ for every input sample $S$, then $\mathcal{H}$ is said to be Privately Agnostically and Properly PAC learnable (PAP-PAC-learnable).

In some of our proofs it will be convenient to consider private learning algorithms whose privacy parameter $\alpha$ satisfies $\alpha \leq 1$ (rather than $\alpha = O(1)$ as in the definition of private algorithms). This can be done without loss of generality due to privacy amplification theorems (see, for example (similar, for example [35] (Definition 8.2) and references within (see also the full version [10] for further details).

**Sanitization.** The notion of sanitization has been introduced by **(author?)** [9] and further studied in [6]. Let $\mathcal{H} \subseteq \{0, 1\}^{\mathcal{W}}$ be a class of functions. An $(\epsilon, \delta, \alpha, \beta, m)$-*sanitizer* for $\mathcal{H}$ is an $(\alpha, \beta)$-private algorithm $M$ that receives as an input a sample $S \in \mathcal{W}^m$ and outputs a function $\text{Est} : \mathcal{H} \to [0, 1]$ such that with probability at least $1 - \delta$,

$$(\forall h \in \mathcal{H}) : \left| \text{Est}(h) - \frac{|\{w \in S : h(w) = 1\}|}{|S|} \right| \leq \epsilon.$$

We say that $\mathcal{H}$ is *sanitizable* if there exists an algorithm $M$ and a bound $m(\epsilon, \delta) = \text{poly}(1/\epsilon, 1/\delta)$ such that for every $\epsilon, \delta > 0$, the restriction of $M$ to samples of any size $m \geq m(\epsilon, \delta)$ is an $(\epsilon, \delta, \alpha, \beta, m)$-sanitizer for $\mathcal{H}$ with $\alpha = \alpha(m) = O(1)$ and $\beta = \beta(m)$ negligible.

**Private Uniform Convergence.** A basic concept in Statistical Learning Theory is the notion of *uniform convergence*. In a nutshell, a class of hypotheses $\mathcal{H}$ satisfies the uniform convergence property if for any unknown distribution $\mathbb{P}$ over examples, one can uniformly estimate the expected losses of all hypotheses in $\mathcal{H}$ given a large enough sample from $P$. Uniform convergence and statistical learning are closely related. For example, the *Fundamental Theorem of PAC Learning* asserts that they are equivalent for binary-classification [33].

This notion extends to the setting of private learning: a class $\mathcal{H}$ satisfies the *Private Uniform Convergence* property if there exists a differentially private algorithm $M$ and a sample complexity bound $m(\epsilon, \delta) = \text{poly}(1/\epsilon, 1/\delta)$ such that for every distribution $\mathbb{P}$ over $\mathcal{W} \times \{0, 1\}$ the following holds: if $M$ is given an input sample $S$ of size at least $m(\epsilon, \delta)$ which is drawn independently from $\mathbb{P}$, then it outputs an estimator $\hat{L} : \mathcal{H} \to [0, 1]$ such that with probability at least $(1 - \delta)$ it holds that

$$(\forall h \in \mathcal{H}) : \big|\hat{L}(h) - L_{\mathbb{P}}(h)\big| \leq \epsilon.$$

Note that without the privacy restriction, the estimator

$$\hat{L}(h) = L_S(h) := \frac{|\{(w_i, y_i) \in S : h(w_i) \neq y_i\}|}{|S|}$$

satisfies the requirement for $m = \tilde{O}(d/\epsilon^2)$, where $d$ is the VC-dimension of $\mathcal{H}$; this follows by the celebrated VC-Theorem [36, 33].

# 3 Problem Setup

We assume a domain $\mathcal{X}$ and we let $\mathcal{D} \subseteq \{0, 1\}^{\mathcal{X}}$ be a class of functions over $\mathcal{X}$. The class $\mathcal{D}$ is referred to as the *discriminating functions class* and its members $d \in \mathcal{D}$ are called *discriminating functions* or *distinguishers*. We let $\Delta(\mathcal{X})$ denote the space of distributions over $\mathcal{X}$. Given two distributions $p, q \in \Delta(\mathcal{X})$, let $\text{IPM}_{\mathcal{D}}(p, q)$ denote the IPM distance between $p$ and $q$ as in Eq. (1).

It will be convenient to assume that $\mathcal{D}$ is *symmetric*, i.e. that whenever $d \in \mathcal{D}$ then also its complement, $1 - d \in \mathcal{D}$. Assuming that $\mathcal{D}$ is symmetric will not lose generality and will help simplify notations. We will also use the following shorthand: given a distribution $p$ and a distinguisher $d$ we will often write

$$p(d) := \mathop{\mathbb{E}}_{x \sim p} [d(x)].$$

Under this assumption and notation we can remove the absolute value from the definition of IPM:

$$\text{IPM}_{\mathcal{D}}(p, q) = \sup_{d \in \mathcal{D}} \left( p(d) - q(d) \right). \tag{2}$$

## 3.1 Synthetic Data Generators

A synthetic data generator (SDG), without additional constraints, is defined as follows

**Definition 1** (SDG). *An SDG, or a fooling algorithm, for $\mathcal{D}$ with sample complexity $m(\epsilon, \delta)$ is an algorithm $M$ that receives as input a sample $S$ of points from $\mathcal{X}$ and parameters $\epsilon, \delta$ such that the following holds: for every $\epsilon, \delta > 0$ and every target distribution $p_{real}$, if $S$ is an independent sample of size at least $m(\epsilon, \delta)$ from $p_{real}$ then*

$$\Pr \left[ \text{IPM}_{\mathcal{D}}(p_{syn}, p_{real}) < \epsilon \right] \geq 1 - \delta,$$

*where $p_{syn} := M(S)$ is the distribution outputted by $M$, and the probability is taken over $S \sim (p_{real})^m$ as well as over the randomness of $M$.*

We will say that a class is *foolable* if it can be fooled by an SDG algorithm whose sample complexity is $\text{poly}(\frac{1}{\epsilon}, \frac{1}{\delta})$. Foolability, without further constraints, comes with the following characterization which is an immediate corollary (or rather a reformulation) of the celebrated VC Theorem ([36]).

Denote by $M_{emp}$ an algorithm that receives a sample $S$ and returns $M_{emp}(S) := p_S$, the empirical distribution over $S$.

**Observation 1** ([36]). *The following statements are equivalent for a class $\mathcal{D} \subseteq \{0, 1\}^{\mathcal{X}}$:*

1. *$\mathcal{D}$ is PAC–learnable.*

2. *$\mathcal{D}$ is foolable.*

3. *$\mathcal{D}$ satisfies the uniform convergence property.*

4. *$\mathcal{D}$ has a finite VC-dimension.*

5. *$M_{emp}$ is a fooling algorithm for $\mathcal{D}$ with sample complexity $m = O(\frac{\log 1/\delta}{\epsilon^2})$.*

Observation 1 shows that foolability is equivalent to PAC-learnability (and in turn to finite VC dimension). We will later see analogous results for DP–Foolability (which is equivalent to differentially private PAC learnability) and Sequential–Foolability (which is equivalent to online learnability).

We now discuss the two fundamental models that are the focus of this work – DP–Foolability and Sequential–Foolability.

## 3.2 DP–Synthetic Data Generators

We next introduce the notion of a DP–synthetic data generator and DP–Foolability. As discussed, DP-SDGs have been the focus of study of several papers [9, 14, 22, 34, 23, 16].

**Definition 2** (DP-SDG). *A DP-SDG, or a DP-fooling algorithm $M$ for a class $\mathcal{D}$ is an algorithm that receives as an input a finite sample $S$ and two parameters $(\epsilon, \delta)$ and satisfies:*

- **Differential Privacy.** *For every $m$, the restriction of $M$ to input samples $S$ of size $m$ is $(\alpha(m), \beta(m))$-differentially private, where $\alpha(m) = O(1)$ and $\beta(m)$ is negligible.*

- **Fooling.** *$M$ fools $\mathcal{D}$: there exists a sample complexity bound $m = m(\epsilon, \delta)$ such that for every target distribution $p_{real}$ if $S$ is a sample of at least $m$ examples from $p_{real}$ then $\mathrm{IPM}_{\mathcal{D}}(p_{syn}, p_{real}) \leq \epsilon$ with probability at least $1 - \delta$, where $p_{syn}$ is the output of $M$ on the input sample $S$.*

We will say in short that a class $\mathcal{D}$ is DP– Foolable if there exists a DP-SDG for the class $\mathcal{D}$ with sample complexity $m = \mathrm{poly}(1/\epsilon, 1/\delta)$.

### 3.3 Sequential–Synthetic Data Generators

We now describe the second model of foolability which, as discussed, is the technical engine behind our proof of equivalence between DP-foolability and DP-learning.

**Sequential-SDGs** A Sequential-SDG can be thought of as a sequential game between two players called the *generator* (denoted by $G$) and the *discriminator* (denoted by $D$). At the beginning of the game, the discriminator $D$ receives the target distribution which is denoted by $p_{real}$. The goal of the generator $G$ is to find a distribution $p$ such that $p$ and $p_{real}$ are $\epsilon$-indistinguishable with respect to some prespecified discriminating class $\mathcal{D}$ and an error parameter $\epsilon > 0$, i.e.

$$\mathrm{IPM}_{\mathcal{D}}(p, p_{real}) \leq \epsilon.$$

We note that both players know $\mathcal{D}$ and $\epsilon$. The game proceeds in rounds, where in each round $t$ the generator $G$ submits to the discriminator a candidate distribution $p_t$ and the discriminator replies according to the following rule: if $\mathrm{IPM}_{\mathcal{D}}(p_t, p_{real}) \leq \epsilon$ then the discriminator replies "WIN" and the game terminates. Else, the discriminator picks $d_t \in \mathcal{D}$ such that $|p_{real}(d_t) - p_t(d_t)| > \epsilon$, and sends $d_t$ to the generator along with a bit which indicates whether $p_t(d_t) > p_{real}(d_t)$ or $p_t(d_t) < p_{real}(d_t)$. Equivalently, instead of transmitting an extra bit, we assume that the discriminator always sends $d_t \in \mathcal{D} \cup (1 - \mathcal{D})$ s.t.

$$p_{real}(d_t) - p_t(d_t) > \epsilon. \tag{3}$$

**Definition 3** (Sequential–Foolability). *Let $\epsilon > 0$ and let $\mathcal{D}$ be a discriminating class.*

1. *$\mathcal{D}$ is called $\epsilon$-Sequential–Foolable if there exists a generator $G$ and a bound $T = T(\epsilon)$ such that $G$ wins any discriminator $D$ with any target distribution $p_{real}$ after at most $T$ rounds.*

2. *The* round complexity *of Sequential–Fooling $D$ is defined as the minimal upper bound $T(\epsilon)$ on the number of rounds that suffice to $\epsilon$–Fool $\mathcal{D}$.*

3. *$\mathcal{D}$ is called Sequential–Foolable if it is $\epsilon$-Sequential foolable for every $\epsilon > 0$ with $T(\epsilon) = \mathrm{poly}(1/\epsilon)$.*

In the next section we will see that if $\mathcal{D}$ is $\epsilon$-Sequential–Foolabe for some fixed $\epsilon < 1/2$ then it is Sequential–Foolable with round complexity $T(\epsilon) = O(1/\epsilon^2)$.

## 4 Results

Our main result characterizes DP–Foolability in terms of basic notions from differential privacy and PAC learning.

**Theorem 1** (Characterization of DP–Fooling). *The following statements are equivalent for a class $\mathcal{D} \subseteq \{0,1\}^X$:*

1. *$\mathcal{D}$ is privately and properly learnable in the agnostic PAC setting.*

2. *$\mathcal{D}$ is DP–Foolable.*

3. *$\mathcal{D}$ is sanitizable.*

4. *$D$ satisfies the private uniform convergence property.*

Theorem 1 shows a qualitative equivalence between the relevant four notions, quantitative bounds on the entailed sample complexity are provided in the full version [10].

The implication Item 3 $\implies$ Item 1 was known prior to this work and was proven in [6] (albeit the pure case). The equivalence among Items 2 to 4 is natural and expected. Indeed, each of them expresses the existence of a private algorithm that *publishes, privately, certain estimates of all functions in $\mathcal{D}$.*

The fact that Item 1 implies the other three items is perhaps more surprising, and the main contribution of this work, and we show that Item 1 implies Item 2. Our proof of that exploits the Sequential framework. In a nutshell, we observe that a class that is both sequentially foolable and privately pac learnable is also DP-foolable: this result follows by constructing a sequential SDG that with a private discriminator, that is assumed to exists, combined with standard compositional and preprocessing arguments regarding the privacy of the generators output.

Thus to prove the implication we only need to show that private PAC learning implies sequential foolability. This result follows from Corollary 2 that provides characterization of sequential foolable classes as well as a recent result by **(author?)** [1] that shows that private PAC learnable classes have finite Littlestone dimension. See the full version [10] for a complete proof.

**Private learnability versus private uniform convergence.** The equivalence Item 1 $\iff$ Item 4 is between private learning and private uniform convergence. The non-private analogue of this equivalence is a cornerstone in statistical learning; it reduces the statistical challenge of minimizing an unknown population loss to an optimization problem of minimizing a known empirical estimate. In particular, it yields the celebrated *Empirical Risk Minimization* (ERM) principle: *"Output $h \in \mathcal{H}$ that minimizes the empirical loss"*. We therefore highlight this equivalence in the following corollary:

**Corollary 1** (Private proper learning = private uniform convergence). *Let $\mathcal{H} \subseteq \{0,1\}^{\mathcal{X}}$. Then $\mathcal{H}$ is privately and properly PAC learnable if and only if $\mathcal{H}$ satisfies the private uniform convergence property.*

**Sequential–SDGs** We next describe our characterization of Sequential-SDGs. As discussed, this characterization is the technical heart behind the equivalence between private PAC learning and DP-foolability. Nevertheless we believe that it may be of interest of its own right. We thus provide quantitative upper and lower bounds on the round complexity of Sequential-SDGs in terms of the Littlestone dimension (see [7] or the full version [10] for the exact definition).

**Theorem 2** (Quantitative round-complexity bounds). *Let $\mathcal{D}$ be a discriminating class with dual Littlestone dimension $\ell^*$ and let $T(\epsilon)$ denote the round complexity of Sequential–Fooling $\mathcal{D}$. Then,*

1. *$T(\epsilon) = O\!\left(\frac{\ell^*}{\epsilon^2} \log \frac{\ell^*}{\epsilon}\right)$ for every $\epsilon$.*

2. *$T(\epsilon) \geq \frac{\ell^*}{2}$ for every $\epsilon < \frac{1}{2}$.*

It would be interesting to close the gap between the two bounds in terms of $\epsilon > 0$, and we leave it for future work.

To prove Item 1 we construct a generator with winning strategy which we outline in the full version [10]. A complete proof of Theorem 2 appears in the full version [10]. As a corollary we get the following characterization of Sequential–Foolability:

**Corollary 2** (Characterization of Sequential–Foolability). *The following are equivalent for $\mathcal{D} \subseteq \{0,1\}^X$:*

1. *$\mathcal{D}$ is Sequential–Foolable.*

2. *$\mathcal{D}$ is $\epsilon$-Sequential–Foolable for some $\epsilon < 1/2$.*

3. *$\mathcal{D}$ has a finite dual Littlestone dimension.*

4. *$\mathcal{D}$ has a finite Littlestone dimension.*

Corollary 2 follows directly from Theorem 2 (which gives the equivalences 1 $\iff$ 2 $\iff$ 3) and from [8] (which gives the equivalence 3 $\iff$ 4, see the full version [10] for further detail).

**Tightness of $\epsilon = \frac{1}{2}$.** The implication Item 2 $\implies$ Item 1 can be seen as a boosting result: i.e. "weak" foolability for some fixed $\epsilon < 1/2$ implies "strong" foolability for every $\epsilon$. The following example demonstrates that the dependence on $\epsilon$ in Item 2 can not be improved beyond $\frac{1}{2}$: let $\mathcal{X}$ be the unit circle in $\mathbb{R}^2$, and let $\mathcal{D}$ consist of all arcs whose length is exactly half of the circumference. It is easy to verify that the uniform distribution $\mu$ over $\mathcal{X}$ satisfies $\mathrm{IPM}_\mathcal{D}(\mu, p_{real}) \leq \frac{1}{2}$ for any target distribution $p_{real}$ (since $\mu(d) = \frac{1}{2}$ for all $d \in \mathcal{D}$). Therefore $\mathcal{D}$ is $(\epsilon = \frac{1}{2})$-Sequential–Foolable with round complexity $T(\frac{1}{2}) = 1$. On the other hand, $\mathcal{D}$ has an infinite Littlestone dimension and therefore is not Sequential–Foolable.

**Sequential-SDGs versus DP-SDGs** So far we have introduced and characterized two formal setups for synthetic data generation. It is therefore natural to compare and seek connections between these two frameworks. We first note that the DP setting may only be more restrictive than the Sequential setting:

**Corollary 3** (DP–Foolability implies Sequential–Foolability)**.** *Let $\mathcal{D}$ be a class that is DP–Foolable. Then $\mathcal{D}$ has finite Littlestone dimension and in particular is Sequential–Foolable.*

Corollary 3 follows from Theorem 1: indeed, the latter yields that DP–Foolability is equivalent to Private agnostic proper -PAC learnability (PAP-PAC), and by [1] PAP-PAC learnability implies a finite Littlestone dimension which by Corollary 2 implies Sequential–Foolability.

**Towards a converse of Corollary 3.** By the above it follows that the family of classes $\mathcal{D}$ that can be fooled by a DP algorithm is contained in the family of all Sequential–Foolable classes; specifically, those which admit a Sequential-SDG with a differentially private discriminator.

We do not know whether the converse holds; i.e. whether "Sequential–Foolability $\implies$ DP– Foolability". Nevertheless, the implication "PAP-PAC learnability $\implies$ DP–Foolability" (Theorem 1) can be regarded as an intermediate step towards this converse. Indeed, as discussed above, PAP-PAC learnablity implies Sequential–Foolablility. It is therefore natural to consider the following question, which is equivalent[2] to the converse of Corollary 3:

**Question 1.** *Let $\mathcal{D}$ be a class that has finite Littlestone dimension. Is $\mathcal{D}$ properly and privately learnable in the agnostic PAC setting?*

A weaker form of this question – Whether every Littlestone class is privately PAC Learnable? – was posed by [1] as an open question (and was recently resolved in [11]).

## 5 Discussion

In this work we develop a theory for two types of constrained-SDG, sequential and private. Let us now discuss SDGs more generally, and we broadly want to consider algorithms that observe data, sampled from some real-life distribution, and in turn generate new synthetic examples that *resemble* real-life samples, without any a-priori constraints. For example, consider an algorithm that receives as input some tunes from a specific music genre (e.g. jazz, rock, pop) and then outputs a new tune.

Recently, there has been a remarkable breakthrough in the the construction of such SDGs with the introduction of the algorithmic frameworks of *Generative Adversarial Networks* (GANs) [18, 17], as well as Variational AutoEncoders (VAE) [26, 31]. In turn, the use of SDGs has seen many potential applications [24, 30, 38]. Here we follow a common interpretation of SDGs as *IPM minimizers* [2, 4]. However, it was also observed [2, 3] that there is a critical gap between the task of generating *new* synthetic data (such as new tunes) and the IPM minimization problem: In detail, Observation 1 shows that the IPM framework allows certain "bad" solutions that *memorize*. Specifically, let $S$ be a sufficiently large independent sample from the target distribution and consider the *empirical distribution* as a candidate solution to the IPM minimization problem. Then, with high probability, the IPM distance between the empirical and the target distribution vanishes as $|S|$ grows.

To illustrate the problem, imagine that our goal is to generate new jazz tunes. Let us consider the discriminating class of all human music experts. The solution suggested above uses the empirical

distribution and simply "generates" a tune from the training set[3]. This clearly misses the goal of generating new and original tunes but the IPM distance minimization framework does not discard this solution. For this reason we often invoke further restrictions on the SDG and consider constrained-SDGs. For example, [4] suggests to restrict the class of possible outputs $p_{syn}$ and shows that, under certain assumptions on the distribution $p_{real}$, the right choice of class $\mathcal{D}$ leads to learning the true underlying distribution (in Wasserstein distance).

In this work we explored two other types of constrained-SDGs, DP–SDGs and Sequential–SDGs, and we characterized the foolable classes in a distribution independent model, i.e. without making assumptions on the distribution $p_{real}$. One motivation for studying these models, as well as the interest in a distribution independent setting, is the following underlying question:

The output of Synthetic Data Generators should be **new** examples. But in what sense we require the output to be novel or *distinct* from the training set? How and in what sense we should avoid copying the training data or even outputting a memorized version of it?

To answer such questions is of practical importance. For example, consider a company that wishes to automatically generate music or images to be used commercially. One approach could be to train an SDG, and then sell the generated output. What can we say about the output of SDGs in this context? Are the images generated by the SDG original? Are they copying the data? or breaching copyright?

In this context, the differentially private setup comes with a very attractive interpretation that provides further motivation to study DP-SDGs, beyond preserving privacy of the dataset. To illustrate our interpretation of differential privacy as a criterion for originality consider the following situation: imagine that Lisa is a learning painter. She has learned to paint by observing samples of painting, produced by a mentor painter Mona. After a learning process, she draws a new painting $L$. Mona agrees that this new painting is a valid work of art, but Mona claims the result is not an original painting but a mere copy of a painting, say $M$, produced by Mona.

How can Lisa argue that paint $L$ is not a plagiary? The easiest argument would be that she had never observed $M$. However, this line of defence is not always realistic as she must observe *some* paintings. Instead, we will argue using the following thought experiment: *What if* Lisa never observed $M$? Might she still create $L$? If we could prove that this is the case, then one could argue similarly that $L$ is not a palgiary.

The last argument is captured by the notion of *differential privacy*. In a nutshell, a randomized algorithm that receives a sequence of data points $\bar{x}$ as input is differentially private if removing/replacing a single data point in its input, does not affect its output $y$ by much; more accurately, for any event $E$ over the output $y$ that has non-negligible probability on input $\bar{x}$, then the probability remains non-negligible even after modifying one data point in $\bar{x}$.

The sequential setting also comes with an appealing interpretation in this context. A remarkable property of existing SDGs (e.g. GANs), that potentially reduces the likeliness of memorization, is that the generator's access to the sample is masked. In more detail, the generator only has restricted access to the training set via feedback from a discriminator that observes real data vs. synthetic data. Thus, potentially, the generator may avoid degenerate solutions that memorize. Nevertheless, even though the generator is not given a direct access to the training data, it could still be that information about this data could "leak" through the feedback it receives from the discriminator. This raises the question of whether Sequential–Foolability can provide guarantees against memorization, and perhaps more importantly, in what sense? To start answering this question part of this work aims to understand the interconnection between the task of Sequential-Fooling and the task of DP–Fooling.

Finally, the above questions also motivate our interest in a distribution-independent setting, that avoids assumptions on the distribution $p_{real}$ which we often don't know. In detail, if we only cared about the resemblence between $p_{real}$ and $p_{syn}$ then we may be content with any algorithm that performs well in practice regardless of whether certain assumptions that we made in the analysis hold or not. But, if we care to obtain guarantees against copying or memorizing, then these should principally hold. And thus we should prefer to obtain our guarantees without too strong assumptions on the distribution $p_{real}$.

## Acknowledgments and Disclosure of Funding

R.L is supported by an ISF grant no. 2188/20 and partially funded by an unrestricted gift from Google. Any opinions, findings, and conclusions or recommendations expressed in this work are those of the author(s) and do not necessarily reflect the views of Google. S.M is supported by the Israel Science Foundation (grant No. 1225/20), by an Azrieli Faculty Fellowship, and by a grant from the United States - Israel Binational Science Foundation (BSF). Part of this work was done while the author was at Google Research.

## Broader Impact

There are no foreseen ethical or societal consequences for the research presented herein.

## Footnotes

[1] I.e. $\beta(m) = o(m^{-k})$ for every $k > 0$.

[2]I.e. an affirmative answer to Question 1 is equivalent to the converse of Corollary 3.

[3]There are at most $7 \cdot 10^9$ music experts in the world. Hence, by standard concentration inequalities a sample of size roughly $\frac{9}{\epsilon^2} \log 10$ suffices to achieve IPM distance at most $\epsilon$ with high probability.

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
