[Supplementary Material]

# Synthetic Data Generators – Sequential and Private

## Abstract

We study the sample complexity of private synthetic data generation over an unbounded sized class of statistical queries, and show that any class that is privately proper PAC learnable admits a private synthetic data generator (perhaps non-efficient). A differentially private synthetic generator is an algorithm that receives a IID data and publishes synthetic data that is indistinguishable from the true data w.r.t a given fixed class of statistical queries. The synthetic data set can then be used by a data scientist without compromising the privacy of the original data set.

Previous work on synthetic data generators focused on the case that the query class $\mathcal{D}$ is finite and obtained sample complexity bounds that scale logarithmically with the size $|\mathcal{D}|$. Here we construct a private synthetic data generator whose sample complexity is independent of the domain size, and we replace finiteness with the assumption that $\mathcal{D}$ is privately PAC learnable (a formally weaker task, hence we obtain equivalence between the two tasks).

Our proof relies on a new type of synthetic data generator, Sequential Synthetic Data Generators, which we believe may be of interest of their own right. A sequential SDG is defined by a sequential game between a generator that proposes synthetic distributions and a discriminator that tries to distinguish between real and fake distributions. We characterize the classes that admits a sequential-SDG and show that they are exactly Littlestone classes. Given the online nature of the Sequential setting, it is natural that Littlestone classes arise in this context. Nevertheless, the characterization of Sequential–SDGs by Littlestone classes turns out to be technically challenging, and to the best of the authors knowledge, does not follow via simple reductions to online prediction.

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

- Let $\mathcal{D}$ be a symmetric class with $\mathrm{Ldim}^*(\mathcal{D}) = \ell^*$, and let $\epsilon > 0$ be the error parameter. Pick $\mathcal{A}$ to be an online learner for the dual class $\mathcal{X}$ like in Corollary 4, and set

$$T = \Big\lceil \frac{4\ell^*}{\epsilon^2} \log \frac{4\ell^*}{\epsilon^2} \Big\rceil = O\Big(\frac{\ell^*}{\epsilon^2} \log \frac{\ell^*}{\epsilon}\Big).$$

- Set $\hat{f}_1(\bar{d}) = \mathbb{E}_{d \sim \bar{d}}[f_1(d)]$ as the predictor of $\mathcal{A}$ at its initial state.
- For $t = 1, \ldots, T$
  1. **If** there exists $p_t \in \Delta(\mathcal{X})$ such that

$$(\forall d \in \mathcal{D}): \underset{x \sim p_t}{\mathbb{E}}\,[f_t(d) - x(d)] \leq \frac{\epsilon}{2},$$

     **then**
     - pick such a $p_t$ and submit it to the discriminator.
       * If the discriminator replies with "Win" then output $p_t$.
       * Else, receive from the discriminator $d_t \in \mathcal{D}$ such that $p_{real}(d_t) - p_t(d_t) \geq \epsilon$
       * Set $\bar{d}_t = \delta_{d_t}$, and $y_t = 1$.
  2. **Else**
     - Find $\bar{d}_t \in \Delta(\mathcal{D})$ such that

$$\big(\forall x \in \mathcal{X}\big): \underset{d \sim \bar{d}_t}{\mathbb{E}}\,[f_t(d) - x(d)] > \frac{\epsilon}{2}$$

       (if no such $\bar{d}_t$ exists then output *"error"*).
     - Set $y_t = 0$.
     - Submit $p_t = p_{t-1}$ to the discriminator and proceed to item 3 below (i.e. here the generator sends a dummy distribution to the discriminator and ignores the answer).
  3. Update $\mathcal{A}$ with the observation $(\bar{d}_t, y_t)$, receive $\hat{f}_{t+1}$, set $f_{t+1}$ such that $\hat{f}_{t+1}(\bar{d}) = \mathbb{E}_{\bar{d}}[f_{t+1}(d)]$ (such $f_{t+1}$ exists by the assumed properties of $\mathcal{A}$ – see Corollary 4), and proceed to the next iteration.
- Output "Lost" (we will prove that this point is never reached).

Figure 1: A fooling strategy for the generator with respect to a symmetric discriminating class $\mathcal{D}$.

## Footnotes

[1]I.e. $\beta(m) = o(m^{-k})$ for every $k > 0$.

[2]I.e. an affirmative answer to Question 1 is equivalent to the converse of Corollary 3.

[3]There are at most $7 \cdot 10^9$ music experts in the world. Hence, by standard concentration inequalities a sample of size roughly $\frac{9}{\epsilon^2} \log 10$ suffices to achieve IPM distance at most $\epsilon$ with high probability.

[4]The same notation will be used for infinite classes also. However we will properly define the the measure space and $\sigma$-algebra at later sections when we extend the results to the infinite regime.

[5]A function $g : \Delta(\mathcal{W}) \to \mathbb{R}$ is *linear* if $g\big(\alpha p_1 + (1 - \alpha)p_2\big) = \alpha g(p_1) + (1 - \alpha)g(p_2)$, for all $\alpha \in [0, 1]$

[6]Note that in order to apply Lemma 3 on $M_1$, we need to assume that $M$ satisfies $(\alpha, \beta)$ privacy with $\alpha \leq 1$. This assumption does not lose generality – see the paragraph following the definition of Private PAC Learning.

[7]In the sense that every other topology with this property contains all open sets in the weak\* topology.

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

# A   Background

## A.1   Primineries

In this section we review some of the basic notations we will use as well as discuss further some standard definitions and notions in differential privacy and online learning.

We continue here the convention of Section 2, and in this section we let $\mathcal{W}$ denote the domain and $\mathcal{H} \subseteq \{0,1\}^{W}$ to denote the functions class.

### A.1.1   Notations

For a finite[4] set $\mathcal{W}$, let $\Delta(\mathcal{W})$ denote the space of probability measures over $\mathcal{W}$. Note that $\mathcal{W}$ naturally embeds in $\Delta(\mathcal{W})$ by identifying $w \in \mathcal{W}$ with the Dirac measure $\delta_w$ supported on $w$. Therefore, every $f : \Delta(\mathcal{W}) \to \mathbb{R}$ induces a $\mathcal{W} \to \mathbb{R}$ function via this identification. In the other

459   direction, every $f : \mathcal{W} \to \mathbb{R}$ naturally extends to a linear[5] map $\hat{f} : \Delta(\mathcal{W}) \to \mathbb{R}$ which is defined
460   by $\hat{f}(p) = \mathbb{E}_p[f]$ for every $p \in \Delta(\mathcal{W})$.

461   We will often deal with boolean functions $f : \mathcal{W} \to \{0, 1\}$, and in some cases we will treat $f$ as
462   the subset of $\mathcal{W}$ that it indicates. For example, given a distribution $p \in \Delta(\mathcal{W})$ we will use $p(f)$ to
463   denote the measure of the subset that $f$ indicates (i.e. $p(f) = \Pr_{w \sim p}[f(w) = 1]$). Given a class of
464   functions $F \subseteq \{0, 1\}^{\mathcal{W}}$, its *dual class* is a class of $F \to \{0, 1\}$ functions, where each function in it is
465   associated with $w \in \mathcal{W}$ and acts on $F$ according to the rule $f \mapsto f(w)$. By a slight abuse of notation
466   we will denote the dual class with $\mathcal{W}$ and use $w(f)$ to denoted the function associated with $w$ (i.e.
467   $w(f) := f(w)$ for every $f \in F$).

468   Given a sample $S = (w_1, \ldots, w_m) \in \mathcal{W}^m$, the *empirical distribution* induced by $S$ is the discrete
469   distribution $p_S$ defined by $p_S(w) = \frac{1}{m} \sum_{i=1}^{m} 1[w = w_i]$.

### A.1.2   Basic properties of Differential Privacy

471   We will use the following three basic properties of algorithmic privacy.

472   **Lemma 1** (Post-Processing (Lemma 2.1 in [41])). *If $M : \mathcal{W}^m \to \Sigma$ is $(\alpha, \beta)$-differentially private*
473   *and $F : \Sigma \to Z$ is any (possibly randomized) function, then $F \circ M : \mathcal{W}^m \to Z$ is $(\alpha, \beta)$-differentially*
474   *private.*

475   **Lemma 2** (Composition (Lemma 2.3 in [41])). *Let $M_1, ..., M_k : \mathcal{W}^m \to \Sigma$ be $(\alpha, \beta)$-differentially*
476   *private algorithms, and define $M : \mathcal{W}^M \to \Sigma^k$ by*

$$M(\Omega) = \big(M_1(\Omega), M_2(\Omega), \ldots, M_k(\Omega)\big).$$

477   *Then, M is $(k\alpha, k\beta)$-differentially private.*

478   **Lemma 3** (Privacy Amplification (Lemma 4.12 in [10])). *Let $\alpha \leq 1$ and let $M$ be a $(\alpha, \beta)$-*
479   *differentially private algorithm operating on databases of size $u$. For $v > 2u$, construct an algorithm*
480   *$M'$ that on input database $\Omega \in \mathcal{W}^v$ subsamples (with replacement) $u$ points from $\Omega$ and runs $M$ on*
481   *the result. Then $M'$ is $(\tilde{\alpha}, \tilde{\beta})$-differentially private for*

$$\tilde{\alpha} = 6\alpha u/v \quad \tilde{\beta} = \exp(6\alpha u/v)\frac{4u}{v}\beta.$$

482   We remark that the requirement $\alpha \leq 1$ can be replaced by $\alpha \leq c$ for any constant $c$ at the expanse of
483   increasing the constant factors in the definitions of $\tilde{\alpha}$ $\tilde{\beta}$. This follows by the same argument that is
484   used to prove Lemma 3 in [10].

### A.1.3   Littlestone Dimension and Online Learning

486   We begin be recalling the basic notion of Littlestone dimension.

487   **Littlestone Dimension**   The Littlestone dimension is a combinatorial parameter that characterizes
488   regret bounds in online learning, but also have recently been related to other concepts in machine
489   learning such as differentially private learning [1]. Perhaps surprisingly, the notion also plays a
490   central role in Model Theory ([39, 12], and see [1] for further discussion).

491   The definition of this parameter uses the notion of *mistake-trees*: these are binary decision trees whose
492   internal nodes are labelled by elements of $\mathcal{W}$. Any root-to-leaf path in a mistake tree can be described
493   as a sequence of examples $(w_1, y_1), ..., (w_d, y_d)$, where $w_i$ is the label of the $i$'th internal node in the
494   path, and $y_i = +1$ if the $(i + 1)$'th node in the path is the right child of the $i$'th node, and otherwise
495   $y_i = 0$. We say that a tree $T$ is *shattered* by $\mathcal{H}$ if for any root-to-leaf path $(w_1, y_1), ..., (w_d, y_d)$ in $T$
496   there is $h \in \mathcal{H}$ such that $h(w_i) = y_i$, for all $i \leq d$.

497   The Littlestone dimension of $\mathcal{H}$, denoted by $\mathrm{Ldim}(\mathcal{H})$, is the maximum depth of a complete tree that
498   is shattered by $\mathcal{H}$.

499   The *dual Littlestone Dimension* which we will denote by $\mathrm{Ldim}^*(\mathcal{H})$ is the Littlestone dimension of
500   the dual class (i.e. we consider $\mathcal{W}$ as the hypothesis class and $\mathcal{H}$ is the domain). We will use the
501   following fact:

**Lemma 4.** *[Corollary 3.6 in [7]] Every class $\mathcal{H}$ has a finite Littlestone dimension if and only if it has a finite dual Littlestone dimension. Moreover we have the following bound:*

$$\mathrm{Ldim}^*(\mathcal{H}) \le 2^{2^{\mathrm{Ldim}(\mathcal{H})+2}} - 2$$

**Online Learning**   The Online learnability of Littlestone classes has been established by [30] in the realizable case and by [6] in the agnostic case. Ben-David et al's [6] agnostic *Standard Online Algorithm* (SOA) will serve as a workhorse for our main results and we thus recall the online learning setting and state the relevant results. For a more exaustive survey on online learning we refer the reader to [11, 38].

In the a binary online setting we assume a domain $\mathcal{W}$ and a space of hypotheses $\mathcal{H} \subseteq \{0,1\}^{\mathcal{W}}$. We consider the following *oblivious* setting which can be described as a repeated game between a learner $L$ and an adversary continuing for $T$ rounds; the *horizon $T$* is fixed and known in advanced to both players. At the beginning of the game, the adversary picks a sequence of labelled examples $(w_t, y_t)_{t=1}^T \subseteq \mathcal{W} \times \{0,1\}$. Then, at each round $t \le T$, the learner chooses (perhaps randomly) a mapping $f_t : \mathcal{W} \to [0,1]$ and then gets to observe the labelled example $(w_t, y_t)$. The performance of the learner $L$ is measured by her *regret*, which is the difference between her loss and the loss of the best hypothesis in $\mathcal{H}$:

$$\mathrm{REGRET}_T(L; \{w_t, y_t\}_{t=1}^T) = \sum_{t=1}^T \mathbb{E}\left[|f_t(w_t) - y_t|\right] - \min_{h \in H} \sum |h(w_t) - y_t|, \qquad (4)$$

where the expectation is taken over the randomness of the learner. Define

$$\mathrm{REGRET}_T(L) = \sup_{\{w_t, y_t\}_{t=1}^T} \mathrm{REGRET}_T(L; \{w_t, y_t\}_{t=1}^T).$$

The following result establishes that Littlestone classes are learnable in this setting:

**Theorem 3.** *[[6]] Let $\mathcal{H}$ be a class with Littlestone dimension $\ell$ and let $T$ be the horizon. Then, there exists an online learning algorithm $L$ such that*

$$\mathrm{REGRET}_T(L) \le \sqrt{\frac{1}{2}\ell \cdot T \log T}$$

We will need the following corollary of Theorem 3. Recall that $\Delta(\mathcal{W})$ denotes the class of distributions over $\mathcal{W}$, and that every $f : \mathcal{W} \to [0,1]$ extends linearly to $\Delta(\mathcal{W})$ by $\hat{f}(p) = \mathbb{E}_{w \sim p}[f(w)]$. The next statement concerns an online setting where the labelled example are of the form $(p_t, y_t) \in \Delta(\mathcal{W}) \times \{0,1\}$, and the regret of a learner $L$ with respect to $\mathcal{H} \subseteq \{0,1\}^{\mathcal{W}}$ is defined by replacing each $h$ by its linear extension $\hat{h}$:

$$\mathrm{REGRET}_T(L; \{p_t, y_t\}_{t=1}^T) = \sum_{t=1}^T \mathbb{E}\left[|f_t(p_t) - y_t|\right] - \min_{h \in H} \sum |\hat{h}(p_t) - y_t|$$

$$= \sum_{t=1}^T \mathbb{E}\left[|f_t(p_t) - y_t|\right] - \min_{h \in H} \sum |\mathbb{E}_{x \sim p_t}[h(w)] - y_t|$$

**Corollary 4.** *Let $\mathcal{H}$ be a finite class with Littlestone dimension $\ell$ and let $T$ be the horizon. Then, there exists a deterministic online learner $L$ that receives labelled examples from the domain $\Delta(\mathcal{W})$ such that*

$$\mathrm{REGRET}_T(L) \le \sqrt{\frac{1}{2}\ell T \log T}$$

*Moreover, at each iteration $t$ the predictor used by $L$ is of the form $\hat{f}_t(p) = \mathbb{E}_{w \sim p}[f_t(w)]$, where $f_t$ is some $\mathcal{W} \to [0,1]$ function.*

Corollary 4 follows from Theorem 3; see Appendix C for a proof.

## B  Proofs

### B.1  Proof of Theorem 2

#### B.1.1  Upper Bound: Proof of Item 1

In this section we prove the upper bound presented in Theorem 2 in the case where $\mathcal{X}$ is finite (and in turn, $\mathcal{D} \subseteq \{0,1\}^{\mathcal{X}}$ is also finite). As discussed though, the bounds will be independent of the domain size. The general case is proven in a similar fashion but is somewhat more delicate. The general proof is then given in Appendix D.

First note that we may assume without loss of generality that $\mathcal{D}$ is symmetric. Indeed, if $\mathcal{D}$ is not symmetric then we may replace $\mathcal{D}$ with $\mathcal{D} \cup (1 - \mathcal{D})$, noting that this does not affect the Sequential game, namely (i) $\mathrm{IPM}_{\mathcal{D}} = \mathrm{IPM}_{\mathcal{D} \cup (1-\mathcal{D})}$ (and so the goal of the generator remains the same), and (ii) the set of distinguishers the discriminator may use remains the same (recall that the discriminator is allowed to use distinguishers from $1 - \mathcal{D}$). Also, one can verify that this modification does not change the dual Lttlestone dimension (i.e. $\mathrm{Ldim}^*(\mathcal{D}) = \mathrm{Ldim}^*(\mathcal{D} \cup (1 - \mathcal{D}))$).

Therefore, we assume $\mathcal{D}$ is a finite symmetric class with dual Littlestone dimension $\ell^*$. The generator used in the proof is depicted in Fig. 1. The generator uses an online learner $\mathcal{A}$ for the dual class $\mathcal{X}$ with domain $\Delta(\mathcal{D})$ as in Corollary 4, where the horizon is set to be $T = \left\lceil \frac{4\ell^*}{\epsilon^2} \log \frac{4\ell^*}{\epsilon^2} \right\rceil$. Let $D$ be an arbitrary discriminator, let $p_{real} \in \Delta(\mathcal{X})$ be the target distribution, and let $\epsilon > 0$ be the error parameter. The proof follows from the next lemma:

**Lemma 5.** *Let $\mathcal{D}$ be a finite set of discriminators, let $f : \mathcal{D} \to [0,1]$, Assume that,*

$$\big(\forall p \in \Delta(\mathcal{X})\big)\big(\exists d \in \mathcal{D}\big) : \mathop{\mathbb{E}}_{x \sim p} [f(d) - x(d)]) > \epsilon/2.$$

*Then:*

$$\big(\exists \bar{d} \in \Delta(\mathcal{D})\big)\big(\forall x \in \mathcal{X}\big) : \mathop{\mathbb{E}}_{d \sim \bar{d}} [f(d) - x(d)] > \epsilon/2.$$

Before proving this lemma, we show how it implies the desired upper bound on the round complexity. We first argue that the algorithm never outputs "error": indeed, since $\mathcal{A}$ only uses predictors of the form $\hat{f}_t(\bar{d}) = \mathbb{E}_{\bar{d}}[f_t]$, Lemma 5 implies that whenever Item 2 in the "For" loop is reached then an appropriate $\bar{d}_t \in \Delta(\mathcal{D})$ exists and therefore the algorithm never outputs "error".

Next, we bound the number of rounds: let $T' \leq T$ be the number of iterations performed when the generator $G$ runs against the discriminator $D$. The only way for the generator to lose is if the "For" loop ends without its winning and $T' = T$. Thus, It suffices to show that $T' < T$. The argument proceeds by showing that the regret of $\mathcal{A}$ in each iteration $t \leq T'$ increases by at least $\epsilon/2$. This, combined with the bound on $\mathcal{A}$'s regret (from Corollary 4) will yield the desired bound.

We begin by analyzing the increase in $\mathcal{A}$'s regret. Let $(\bar{d}_1, y_1), \ldots, (\bar{d}_{T'}, y_{T'})$ and $\hat{f}_1, \ldots, \hat{f}_{T'}$ be the sequences obtained during the execution of the algorithm as defined in Fig. 1. Recall from Corollary 4 that $\hat{f}_t(\bar{d}) = \mathbb{E}_{d \sim \bar{d}}[f_t(d)]$, where $f_t : \mathcal{D} \to [0,1]$. We claim that the following holds:

$$(\forall t \leq T') : \begin{cases} \mathbb{E}_{d \sim \bar{d}_t} \big[p_{real}(d) - f_t(d)\big] \geq \frac{\epsilon}{2} & \text{if } y_t = 1, \\ \mathbb{E}_{d \sim \bar{d}_t} \big[f_t(d) - p_{real}(d)\big] \geq \frac{\epsilon}{2} & \text{if } y_t = 0. \end{cases} \tag{5}$$

Indeed, if $y_t = 1$ then by Fig. 1, the chosen $p_t$ satisfies

$$(\forall d \in \mathcal{D}) : f_t(d) - \mathop{\mathbb{E}}_{x \sim p_t} [x(d)] \leq \frac{\epsilon}{2}.$$

Since the discriminator replies with $d_t$ such that $p_{real}(d_t) - p_t(d_t) \geq \epsilon$, and $\bar{d}_t = \delta_{d_t}$, it follows that

$$
\begin{aligned}
\mathop{\mathbb{E}}_{d \sim \bar{d}_t} \big[p_{real}(d) - f_t(d)\big] &= \mathop{\mathbb{E}}_{d \sim \bar{d}_t} [p_{real}(d_t)] - \mathop{\mathbb{E}}_{d \sim \bar{d}_t} [f_t(d_t)] \\
&= p_{real}(d_t) - f_t(d_t) && \text{(because } \bar{d}_t = \delta_{d_t}) \\
&\geq \mathop{\mathbb{E}}_{x \sim p_{real}} [x(d_t)] - \left( \mathop{\mathbb{E}}_{x \sim p_t} [x(d_t)] + \epsilon/2 \right) \\
&= p_{real}(d_t) - (p_t(d_t) + \epsilon/2) \\
&\geq \frac{\epsilon}{2},
\end{aligned}
$$

which is the first case in Eq. (5). Next consider the case when $y_t = 0$. Since the algorithm never outputs "error", Fig. 1 implies that:

$$\left(\forall x \in \mathcal{X}\right) : \hat{f}_t(\bar{d}_t) - \mathbb{E}_{d\sim\bar{d}_t}[x(d)] > \frac{\epsilon}{2}.$$

Therefore, by linearity of expectation, $\mathbb{E}_{d\sim\bar{d}_t}\big[f_t(d) - p_{real}(d)\big] = \hat{f}_t(\bar{d}_t) - \mathbb{E}_{d\sim\bar{d}_t}[p_{real}(d)] \geq \frac{\epsilon}{2}$, which amounts to the second case in Eq. (5).

We are now ready to conclude the proof by showing that $T' < T$. Assume towards contradiction that $T' = T$. Therefore, by Eq. (5):

$$T\frac{\epsilon}{2} \leq \sum_{t=1}^{T}\Big|\mathbb{E}_{d\sim\bar{d}_t}\big[p_{real}(d) - f_t(d)\big]\Big|$$

$$= \sum_{t=1}^{T}\Big|y_t - \mathbb{E}_{d\sim\bar{d}_t}[f_t(d)]\Big| - \Big|y_t - \mathbb{E}_{d\sim\bar{d}_t}[p_{real}(d_t)]\Big|$$

$$\left(y_t = 1 \iff \mathbb{E}_{d\sim\bar{d}_t}[p_{real}(d_t)] \geq \mathbb{E}_{d\sim\bar{d}_t}[f_t(d)]\right)$$

$$= \sum_{t=1}^{T}\Big|y_t - \hat{f}_t(\bar{d}_t)\Big| - \mathbb{E}_{x\sim p_{real}}\left[\Big|y_t - \mathbb{E}_{d\sim d_t}x(d_t)\Big|\right]$$

$$\leq \sum_{t=1}^{T}|y_t - f_t(\bar{d}_t)| - \min_{x\in\mathcal{X}}|y_t - \mathbb{E}_{d\sim d_t}[x(d)]|$$

$$\leq \text{REGRET}_T(\mathcal{A}).$$

$$\leq \sqrt{\frac{1}{2}\ell^* T \log T}$$

Thus, we obtain that $\frac{T}{\log T} \leq \frac{2\ell^*}{\epsilon^2}$, however our choice of $T = \left\lceil \frac{4\ell^*}{\epsilon^2}\log\frac{4\ell^*}{\epsilon^2}\right\rceil$ ensures that this is impossible. Indeed:

$$\frac{T}{\log T} \geq \frac{\frac{4\ell^*}{\epsilon^2}\log\frac{4\ell^*}{\epsilon^2}}{\log\frac{4\ell^*}{\epsilon^2} + \log\log\frac{4\ell^*}{\epsilon^2}}$$

$$= \frac{\frac{4\ell^*}{\epsilon^2}}{1 + \frac{\log\log\frac{4\ell^*}{\epsilon^2}}{\log\frac{4\ell^*}{\epsilon^2}}}$$

$$> \frac{\frac{4\ell^*}{\epsilon^2}}{2}$$

$$= \frac{2\ell^*}{\epsilon^2}.$$

This finishes the proof of Item 1.

We end this section by proving Lemma 5.

***Proof of Lemma 5.*** The proof hinges on Von Neuman's Minimax Theorem. Let $D, f$ as in the formulation of the theorem, and consider the following zero-sum game: the pure strategies of the maximizer are indexed by $d \in \mathcal{D}$, the pure strategies of the minimizer are indexed by $x \in X$, and the payoff (for pure strategies) is defined by $m(d, x) = f(d) - x(d)$. Note that the payoff function for mixed strategies $\bar{d} \in \Delta(\mathcal{D}), p \in \Delta(\mathcal{X})$ satisfies

$$m(\bar{d}, p) = \mathbb{E}_{x\sim p}[\hat{f}(\bar{d}) - \mathbb{E}_{d\sim\bar{d}}x(d)] = \mathbb{E}_{d\sim\bar{d}}\big[f(d) - \mathbb{E}_{x\sim p}[x(d)]\big].$$

We next apply Von Neuman's Minimax Theorem on this game (Here we use the assumption that $\mathcal{X}$ and, in turn, $\mathcal{D}$ are finite). The premise of the lemma amounts to

$$\min_{p\in\Delta(\mathcal{X})} \max_{d\in\mathcal{D}} m(d, p) > \epsilon/2.$$

583 Therefore, by the Minimax Theorem also

$$\max_{\bar{d} \in \Delta(\mathcal{D})} \min_{x \in \mathcal{X}} m(\bar{d}, x) > \epsilon/2,$$

584 which amounts to the conclusion of the lemma. $\qquad\square$

585 **A remark.** A natural variant of the Sequential setting follows by letting the discriminator $D$ to
586 adaptively change the target distribution $p_{real}$ as the game proceeds ($D$ would still be required
587 to maintain the existence of a distribution $p_{real}$ which is consistent with all of its answers). This
588 modification allows for stronger discriminators and therefore, potentially, for a more restrictive notion
589 of Sequential–Foolability. However, the above proof extends to this setting verbatim.

### B.1.2 Lower Bound: Proof of Item 2

591 Let $\mathcal{D}$ be a class as in the theorem statement, let $G$ be a generator for $\mathcal{D}$, and let $\epsilon < \frac{1}{2}$. We will
592 construct a discriminator $D$ and a target distribution $p_{real}$ such that $G$ requires at least $\frac{\ell^*}{2}$ rounds in
593 order to find $p$ such that $\mathrm{IPM}_{\mathcal{D}}(p, p_{real}) \leq \epsilon$.

594 To this end, pick a shattered mistake-tree $\mathcal{T}$ of depth $\ell^*$ whose internal nodes are labelled by elements
595 of $\mathcal{D}$ and whose leaves are labelled by elements of $\mathcal{X}$.

596 **The discriminator.** The target distribution will be a Dirac distribution $\delta_x$ where $x$ is one of the
597 labels of $\mathcal{T}$'s leaves. We will use the following discriminator $D$ which is defined whenever $p_{real}$ is
598 one of these distributions: assume that $p_{real} = \delta_x$, and consider all functions in $\mathcal{D}$ that label the path
599 from the root towards the leaf whose label is $x$,

$$d_1, d_2, \ldots, d_{\ell^*}.$$

600 Let $p_1$ be the distribution the generator submitted in the first round. Then the discriminator picks the
601 first $i$ such that $|p_t(d_1) - p_{real}(d_1)| > \epsilon$, and sends the generator either $d_i$ or $1 - d_i$ according to the
602 convention in Eq. (3). If no such $d_i$ exists, the discriminator outputs WIN. Similarly, at round $t$ let
603 $i_{t-1}$ denote the index of the distinguisher sent in the previous round; then, the discriminator acts the
604 same with the modification that it picks the first $i_{t-1} + 1 \leq i \leq \ell^*$ such that $|p_t(d_i) - p_{real}(d_i)| > \epsilon$.

605 **Analysis.** The following claim implies that for every generator $G$, there exists a distribution $\delta_x$
606 such that if $p_{real} = \delta_x$ then the above discriminator $D$ forces $G$ to play at least $\ell^*/2$ rounds.

607 **Claim 1.** *Let $G$ be a generator for $\mathcal{D}$. Pick $p_{real}$ uniformly at random from the set $\{\delta_x :$*
608 *$x$ labels a leaf in $\mathcal{T}\}$. Then the expected number of rounds in the Sequential game when $G$ is the*
609 *generator and $D = D(\mathcal{T})$ is the discriminator is at least $\frac{\ell^*}{2}$.*

610 *Proof.* For every $i \leq \ell^*$, let $X_i$ denote the indicator of the event that the $i$'th function on the path
611 towards the leaf corresponding to $p_{real}$ was used by $D$ as a distinguisher. Note that the number of
612 rounds $X$ satisfies $X = \sum_{i=1}^{\ell^*} X_i$. Thus, by linearity of expectation it suffices to argue that

$$\mathbb{E}[X_i] = \Pr[X_i = 1] \geq \frac{1}{2}.$$

613 Consider $X_1$: let $p_1$ denote the first distribution submitted by $G$. Note that $X_1 = 1$ if

614     (i) $p_1(d_1) \geq \frac{1}{2}$ and the leaf labelled $x$ belongs to the left subtree from the root, or

615     (ii) $p_1(d_1) < \frac{1}{2}$ and the leaf labelled $x$ belongs to the right subtree from the root.

616 In either way $\Pr[X_1 = 1] \geq \frac{1}{2}$, since this leaf is drawn uniformly. Similarly, for every conditioning on
617 the values of $X_1, \ldots, X_{i-1}$ we have $\Pr[X_i = 1 | X_1 \ldots X_{i-1}] \geq \frac{1}{2}$ (follows from the same argument
618 applied on subtrees corresponding to the conditioning). This yields that $\mathbb{E}[X_i] = \Pr[X_i = 1] \geq \frac{1}{2}$
619 for every $i$ as required.

620     $\qquad\square$

 **B.2 Proof of Theorem 1**

622 **Proof Roadmap.** We will show the following entailments: $1 \Rightarrow 2 \Rightarrow 3 \Rightarrow 1$. Then, given the equiva-
623 lence between Items 1 to 3 we will show that $1 \Leftrightarrow 4$. This will conclude the proof.

624 **Overview of $1 \Rightarrow 2$.** We next overview the derivation of $1 \Rightarrow 2$ which is the most involved derivation.
625 Let $p_{real}$ denote the target distribution we wish to fool. The argument relies on the following simple
626 observation: let $S$ be a sufficiently large independent sample from $p_{real}$. Then, it suffices to privately
627 output a distribution $p_{syn}$ such that $\mathrm{IPM}_{\mathcal{D}}(p_{syn}, p_S) \leq \frac{\epsilon}{2}$, where $p_S$ is the empirical distribution.
628 Indeed, if $S$ is sufficiently large then by standard uniform convergence bounds: $\mathrm{IPM}_{\mathcal{D}}(p_S, p_{real}) \leq \frac{\epsilon}{2}$,
629 which implies that $\mathrm{IPM}_{\mathcal{D}}(p_{syn}, p_{real}) \leq \epsilon$ as required.

630 The output distribution $p_{syn}$ is constructed using a carefully tailored Sequential-SDG with a *private*
631 *discriminator* $D$. That is, $D$'s input distribution is the empirical distribution $p_S$, and for every
632 submitted distribution $p_t$, it either replies with a discriminating function $d_t$ or with "WIN" if no
633 discriminating function exists. The crucial point is that it does so in a differentially private manner
634 with respect to the input sample $S$. The existence of such a discriminator $D$ follows via the assumed
635 PAP-PAC learner.

636 Once the private discriminator $D$ is constructed, we turn to find a generator $G$ with a bounded round
637 complexity. This follows from Theorem 2 and a result by [1, 10]: by [1, 10] PAP-PAC learnability
638 implies a finite Littlestone dimension, and therefore by Theorem 2 there is a generator $G$ with a
639 bounded round complexity. The desired DP fooling algorithm then follows by letting $G$ and $D$ play
640 against each other and outputting the final distribution that $G$ obtains. The privacy guarantee follows
641 by the *composition lemma* (Lemma 2) which bounds the privacy leakage in terms of the number of
642 rounds (which is bounded by the choice of $G$) and the privacy leakage per round (which is bounded
643 by the choice of $D$).

644 One difficulty that is handled in the proof arises because the discriminator is differentially private
645 and because the PAP-PAC algorithm may err with some probability. Indeed, these prevent $D$ from
646 satisfying the requirements of a discriminator as defined in the Sequential setting. In particular, $D$
647 cannot reply deterministically whether $\mathrm{IPM}_{\mathcal{D}}(p_S, p_t) < \epsilon$ as this could compromise privacy. Also,
648 whenever the assumed PAP-PAC algorithm errs, $D$ may reply with an illegal distinguisher that does
649 not satisfy Eq. (3).

650 To overcome this difficulty we ensure that $D$ satisfies the following with high probability: if
651 $\mathrm{IPM}_{\mathcal{D}}(p_S, p_t) > \epsilon$ then $D$ outputs a legal $d_t$, and if $\mathrm{IPM}_{\mathcal{D}}(p_S, p_t) < \frac{\epsilon}{2}$ then it outputs WIN
652 as required. When $\frac{\epsilon}{2} \leq \mathrm{IPM}_{\mathcal{D}}(p_S, p_t) \leq \epsilon$ it may either output WIN or a legal discriminator $d_t$. As
653 we show in the proof, this behaviour of $D$ will not affect the correctness of the overall argument.

654 *Proof of Theorem 1.* As discussed, the equivalence is proven by showing: $1 \Rightarrow 2 \Rightarrow 3 \Rightarrow 1$ and $1 \Leftrightarrow 4$.

655 **$1 \Rightarrow 2$.** Let $p_{real}$ denote the unknown target distribution and let $\epsilon_0, \delta_0$ be the error and confidence
656 parameters. Draw independently from $p_{real}$ a sufficiently large input sample $S$ of size $|S|$ to be
657 specified later. At this point we require $|S|$ to be large enough so that $\mathrm{IPM}_{\mathcal{D}}(p_{real}, p_S) \leq \frac{\epsilon_0}{2}$ with
658 probability at least $1 - \frac{\delta_0}{2}$. By standard uniform convergence bounds ([42]) it suffices to require

$$|S| \geq \Omega\Big(\frac{d + \log(1/\delta_0)}{\epsilon_0^2}\Big), \tag{6}$$

659 where $d$ is the VC-dimension of $\mathcal{D}$ (observe that $\mathcal{D}$ must have a finite VC dimension as it is PAC
660 learnable). By the triangle inequality, this reduces our goal to privately output a distribution $p_{syn}$
661 so that $\mathrm{IPM}_{\mathcal{D}}(p_S, p_{syn}) \leq \frac{\epsilon_0}{2}$ with probability $1 - \frac{\delta_0}{2}$ (this will imply that $\mathrm{IPM}_{\mathcal{D}}(p_{real}, p_{syn}) \leq \epsilon_0$
662 with probability $1 - \delta_0$).

663 As explained in the proof outline, the latter task is achieved by a Sequential-SDG which we will
664 next describe. Inorder to construct the desired Sequential-SDG, we first observe that $\mathcal{D}$ is Sequential–
665 Foolable. Indeed, by Corollary 2 it suffices to argue that $\mathcal{D}$ has a finite Littlestone dimension, which
666 follows by [1] since $\mathcal{D}$ is privately learnable.

667 Now, pick a generator $G$ that fools $\mathcal{D}$ with round complexity $T(\epsilon)$ as in Theorem 2, and pick a
668 discriminator $D$ as in Fig. 2. Note that $D$ uses a PAP-PAC learner for the class $\mathcal{D} \cup (1 - \mathcal{D})$ whose

existence follows from the PAP-PAC learnability of $\mathcal{D}$ via standard arguments (which we omit). The next lemma summarizes the properties of $D$ that are needed for the proof.

**Lemma 6.** *Let $D$ be the discriminator defined in Fig. 2 with input parameters $(\epsilon, \delta, \tau)$ and input sample $S$, and let $M$ be the assumed PAP-PAC learner for $\mathcal{D} \cup (1 - \mathcal{D})$ with sample complexity $m(\epsilon, \delta)$ and privacy parameters $(\alpha, \beta)$. Then, $D$ is $\left(6\tau\alpha(\tau|S|) + \tau, 4e^{6\tau\alpha(\tau|S|)}\tau\beta(\tau|S|)\right)$-private, and if $S$ satisfies*

$$|S| \geq \max\left(\frac{m(\epsilon/8, \tau\delta/2)}{\tau}, \frac{64\log(\tau\delta/2)}{\epsilon\tau}\right) \tag{7}$$

*then following holds with probability at least $(1 - \tau\delta)$*

    *(i) If $D$ outputs $d_t$ then $p_S(d_t) - p_t(d_t) \geq \frac{\epsilon}{2}$.*

    *(ii) If $D$ outputs "WIN" then $\mathrm{IPM}_{\mathcal{D}}(p_S, p_t) \leq \epsilon$.*

We first use Lemma 6 to conclude the proof of $1 \Rightarrow 2$ and then prove Lemma 6.

The fooling algorithm we consider proceeds as follows.

- Set $G$ to be a generator with round complexity $T(\epsilon)$ and set its error parameter to be $\frac{\epsilon_0}{2}$.

- Set the number of rounds $T_0 = \min\{|S|^{0.99}, T(\epsilon_0/4)\}$, and let $\tau_0 = 1/T_0$.

- Set $D$ be the discriminator depicted in Fig. 2 and set its parameters to be $(\epsilon, \delta, \tau) = (\frac{\epsilon_0}{2}, \frac{\delta_0}{2}, \tau_0)$ and its input sample to be $S$.

- Let $G$ and $D$ to play against each other for (at most) $T_0$ rounds.

- Output the final distribution which is held by $G$.

We next prove the privacy and fooling properties as required by a DP algorithm:

**Privacy.** We argue that the algorithm is $(\alpha', \beta')$–private, with $\alpha'(|S|) = O(1)$ and $\beta'(|S|)$ negligible. Note that since $G$ is deterministic then the output distribution $p_{out}$ is completely determined by the sequence of discriminating functions $d_1, \ldots, d_{T'}$ outputted by the discriminator.

For simplicity and without loss of generality we assume that $T' = T_0$: indeed, if $T' < T_0$ then extend it by repeating the last discriminating function; this does not change the fact that $p_{out}$ is determined by the sequence $d_1, \ldots, d_{T'}, \ldots d_{T_0}$.

Recall that by Lemma 6 $D$ is $\left((6\tau_0\alpha(\tau_0|S|) + \tau_0), \left(4e^{6\tau_0\alpha(\tau_0|S|)}\tau_0\beta(\tau_0|S|)\right)\right)$-private. Therefore, since the number of rounds in which $D$ is applied is $T_0$, by *composition* (Lemma 2) and *post-processing* (Lemma 1) it follows that the entire algorithm is

$$\left(T_0\left(6\tau_0\alpha(\tau_0|S|) + \tau_0\right), T_0\left(4e^{6\tau_0\alpha(\tau_0|S|)}\tau_0\beta(\tau_0|S|)\right)\right)\text{-private.}$$

Our choices of $\tau_0 = \frac{1}{T_0}$ and $T_0$ guarantee that $1/\tau_0 < m^{0.99}$, and plugging it in yields privacy guarantee of $(6\alpha(|S|^{0.001}) + 1, 4e^{O(1)}\beta(|S|^{0.001}))$. As $\alpha(|S|^{0.001}) = O(1)$ and $\beta(|S|^{0.001})$ is negligible, the desired privacy guarantee follows.

**Fooling.** First note that if $S$ satisfies Eq. (7) with $(\epsilon, \delta, \tau) := (\epsilon_0, \frac{\delta_0}{2}, \tau_0)$ then with probability at least $1 - \frac{\delta_0}{2}$ the following holds: in every iteration $t \leq T_0$, either $p_S(d_t) - p_t(d_t) \geq \frac{\epsilon_0}{4}$, or the discriminator yields WIN and $\mathrm{IPM}_{\mathcal{D}}(p_S, p_t) \leq \frac{\epsilon_0}{2}$. This follows by a union bound via the utility guarantee in Lemma 6. Assuming this event holds, we claim that if $|S|$ is set to satisfy $|S|^{0.99} \geq T(\frac{\epsilon_0}{4})$ then the output distribution $p_{syn}$ satisfies $\mathrm{IPM}_{\mathcal{D}}(p_S, p_{syn}) \leq \frac{\epsilon_0}{2}$. This follows since as long as the sequential game proceeds the generator suffers a loss of at least $\frac{\epsilon_0}{4}$ in every round, and the number of rounds is set as, in this case, to be $T(\frac{\epsilon_0}{4})$. Therefore we require

$$|S|^{0.99} \geq T\left(\frac{\epsilon_0}{4}\right) = \Omega\left(\frac{\ell^*}{\epsilon_0^2}\log\frac{\ell^*}{\epsilon_0}\right). \tag{8}$$

To conclude, if $|S|$ is set to satisfy Eqs. (6) to (8) then with probability at least $1 - \delta_0$ both $\mathrm{IPM}_{\mathcal{D}}(p_{real}, p_S) \leq \frac{\epsilon_0}{2}$ and $\mathrm{IPM}_{\mathcal{D}}(p_S, p_{syn}) \leq \frac{\epsilon_0}{2}$, which implies that $\mathrm{IPM}_{\mathcal{D}}(p_{real}, p_{syn}) \leq \epsilon_0$ as required. This concludes the proof of 1⇒2.

***Proof of Lemma 6.*** Let $S$ be the input sample, let $p_S$ denote the uniform distribution over $S$, and let $p_t$ denote the distribution submitted by the generator. The discriminator operates as follows (see Fig. 2): it feeds the assumed PAP-PAC learner a labeled sample $S_\ell = \{(x_i, y_i)\}$ that is drawn from the following distribution $q_t$: first the label $y_i$ is drawn uniformly from $\{0, 1\}$; if $y_i = 0$ then draw $x_i \sim p_S$ and if $y_i = 1$ then draw $x_i \sim p_t$. Let $d_t$ denote the output of the PAP-PAC learner on the input sample $S$. Observe that the loss $L_{q_t}(\cdot)$ satisfies

$$L_{q_t}(d) = \frac{p_S(d) + (1 - p_t(d))}{2} = \frac{1 + p_S(d) - p_t(d)}{2}. \tag{9}$$

Next, the discriminator checks whether $p_S(d_t) - p_t(d_t) > \frac{\epsilon}{2}$ (equivalently, if $L_{q_t}(d_t) < \frac{1 - \epsilon/2}{2}$), and sends $d_t$ the generator if so, and reply with "WIN" otherwise. The issue is that checking this "If" condition naivly may violate privacy, and in order to avoid it we add noise to this check by a mechanism from [14] (see Fig. 3): roughly, this mechanism receives a data set of scalars $\Sigma = \{\sigma_i\}_{i=1}^m$, a threshold parameter $c$ and a margin parameters $N$, and outputs $\top$ if $\sum_{i=1}^m \sigma_i > c + O(1/N)$ or $\bot$ if $\sum_{i=1}^m \sigma_i < c - O(1/N)$. The distinguisher applies this mechanism over the sequence of scalars $\{d_t(x_1), \ldots, d_t(x_m)\}$.

We next formally establish the privacy and utility guarantees of $D$. In what follows, assume that the input sample $S$ satisfies Eq. (7),

**Privacy.** The discriminator $D$ is a composition of two procedures, $M_1$ and $M_2$, where $M_1$ applies the PAP-PAC learner $M$ on the random subsample $S_\ell$, and $M_2$ runs the procedure THRESH. Thus, the privacy guarantee will follow from the composition lemma (Lemma 2) if we show that $M_1$ is $(6\tau\alpha(\tau m), 4e^{6\tau\alpha(\tau m)}\tau\beta(\tau m))$-private and $M_2$ is $(\tau, 0)$-private. The privacy guarantee of $M_1$ follows by applying[6] Lemma 3 with $v := |S|$ and $n := |S_\ell| = \tau|S|$, and the privacy guarantee of $M_2$ follows from the statement in Fig. 3 since $\frac{N}{|\Sigma|} = \frac{|S_\ell|}{|S|} = \tau$.

**Utility.** Let $q_t$ denote the distribution from which the subsample $S_\ell$ is drawn. Note that by Eq. (7), $S_\ell = \tau \cdot |S| \geq m(\epsilon/8, \tau\delta/2)$. Therefore, since $M$ PAC learns $\mathcal{D}$, its output $d_t$ satisfies:

$$L_{q_t}(d_t) \leq \min_{d \in \mathcal{D} \cup (1 - \mathcal{D})} L_{q_t}(d) + \frac{\epsilon}{8},$$

with probability at least $1 - \tau\delta/2$. By Eq. (9) this is equivalent to

$$p_S(d_t) - p_t(d_t) \geq \max_{d \in \mathcal{D} \cup (1 - \mathcal{D})} \big( p_S(d) - p_t(d) \big) - \epsilon/4. \tag{10}$$

Now, by plugging in the statement in Fig. 3: $(\Sigma, c, N) := (\{d_t(x)\}_{x \in S}, p_t(d_t) + \frac{5\epsilon}{8}, |S_\ell|)$, and $\gamma := \tau\delta/2$ and conditioning on the event that both $M$ and THRESH succeed (which occurs with probability at least $1 - \tau\delta$) it follows that

(i) If $D$ outputs $d_t$ then

$$p_S(d_t) \geq c - \frac{8\log(1/\gamma)}{N} = p_t(d_t) + \frac{5\epsilon}{8} - \frac{8\log(\tau\delta/2)}{\tau|S|} \geq p_t(d_t) + \frac{\epsilon}{2},$$

where in the last inequality we used that $|S| \geq \frac{64\log(\tau\delta/2)}{\epsilon\tau}$ (by Eq. (7)).

(ii) If $D$ outputs WIN then by a similar calculation $p_S(d_t) \leq p_t(d_t) + \frac{3\epsilon}{4}$ and therefore

$$\mathrm{IPM}_{\mathcal{D}}(p_S, p_t) = \max_{d \in \mathcal{D} \cup (1 - \mathcal{D})} \big( p_S(d) - p_t(d) \big) \leq p_S(d_t) - p_t(d_t) + \frac{\epsilon}{4} \leq \epsilon,$$

where in the first inequality we used Eq. (10).

This concludes the proof of Lemma 6.

$\square$

- Let $M$ be a PAP–PAC learner for the class $\mathcal{D} \cup (1 - \mathcal{D})$ with sample complexity $m(\epsilon, \delta)$.

- Let $\epsilon, \delta, \tau$ be the input parameters.

- Let $S$ be the input sample, let $p_S$ be the uniform distribution over $S$, and let $p_t$ be the distribution submitted by the generator.

- Draw a labelled sample $S_\ell = \{(x_i, y_i)\}$ of size $\tau \cdot |S|$ independently as follows: draw the label $y_i$ uniformly from $\{0, 1\}$

  (i) if $y_i = 0$ then draw $x_i \sim p_S$,
  (ii) if $y_i = 1$ then draw $x_i \sim p_t$.

- Apply the learner $M$ on the sample $S_\ell$ and set $d_t \in \mathcal{D}$ as its output.

- Compute $Z := \mathrm{THRESH}\left(\{d_t(x)\}_{x \in S}, p_t(d_t) + \frac{5\epsilon}{8}, |S_\ell|\right)$.

  (i) If $Z = \top$ then send the generator with $d_t$,
  (ii) else, $Z = \bot$ and reply the generator with "Win".

Figure 2: Depiction of the private discriminator used in Theorem 1. The discriminator holds the target distribution $p_S$, where $S$ is a sufficiently large sample from $p_{real}$. In each round the discriminator decides whether $p_S$ is indistinguishable from the distribution submitted by the generator and replies accordingly.

THRESH. The procedure THRESH receives as input a dataset of scalars $\Sigma = \{\sigma_i\}$, a threshold parameter $c > 0$ and a margin parameter $N$ and has the following properties (see Theorem 3.23 in [14] for proof of existence):

- $\mathrm{THRESH}(\Sigma, c, N)$ is $(N/|\Sigma|, 0)$-private.

- For every $\gamma > 0$:

  - If $\frac{1}{|\Sigma|} \sum_{\sigma \in \Sigma} \sigma > c + \frac{8 \log 1/\gamma}{N}$ then THRESH outputs $\top$ with probability at least $1 - \gamma$
  - If $\frac{1}{|\Sigma|} \sum_{\sigma \in \Sigma} \sigma < c - \frac{8 \log 1/\gamma}{N}$ then THRESH outputs $\bot$ with probability at least $1 - \gamma$

Figure 3: The procedure: THRESH

**2⇒3.** This follows directly from the definition of a DP–Fooling algorithm. Indeed, given a DP–Fooling algorithm with sample complexity $m(\epsilon, \delta)$ and a sample $S$ outputs a distribution $p_{syn}$ such that $\mathrm{IPM}_{\mathcal{D}}(p_{syn}, p_S) \leq \epsilon$, with probability at least $(1 - \delta)$ and satisfies $(\alpha, \beta)$-privacy, with $\alpha = O(1)$ and $\beta$ negligible. To obtain a sanitizer, output the estimate $\mathrm{EST} : \mathcal{D} \to [0, 1]$, where $\mathrm{Est}(d) = \mathbb{E}_{x \sim p_{syn}}[d(x)]$.

**3⇒1.** This follows from Theorem 5.5 in [5].

**4⇒1.** This is an immediate corollary of post-processing for differential privacy (Lemma 1). Indeed, by the private uniform convergence property we can privately estimate the losses of all hypotheses in $\mathcal{D}$, and then output any hypothesis in $\mathcal{D}$ that minimizes the estimated loss.

**1⇒4.** Suppose $\mathcal{D}$ is PAP–PAC learnable by an algorithm $A$. For every function $d \in \mathcal{D}$, let $d'$ denote the $(X \times \{0, 1\}) \to \{0, 1\}$ function defined by $d'((x, y)) = \mathbf{1}[d(x) \neq y]$, and let $\mathcal{D}' = \{d' : d \in \mathcal{D}\}$. Observe that for every sample $S \subseteq (X \times \{0, 1\})^m$:

$$L_S(d) = p_S(d'), \tag{11}$$

where $L_S(d)$ denotes the empirical loss of $d$ and $p_S$ denotes the empirical measure of $d'$.

We claim that $\mathcal{D}'$ is also PAP–PAC learnable: for a $\mathcal{D}'$-example $z' = ((x, y), y')$ let $z$ denote the $\mathcal{D}$-example $(x, |y' - y|)$, and note that $d'$ errs on $z'$ if and only if $d$ errs on $z$. Therefore, a PAP–PAC

757 learner for $\mathcal{D}'$ follows by using this transformation to convert the $\mathcal{D}'$-input sample $S' = \{z_i'\}_{i=1}^m$ to a
758 $\mathcal{D}$ input sample $S = \{z_i\}_{i=1}^m$, applying $A$ on $S$ and outputting $d'$, where $d = A(S)$.

759 Therefore, by $1 \implies 3$ it follows that $\mathcal{D}'$ is sanitizable by a sanitizer $M$ with sample complex-
760 ity $m_1(\epsilon, \delta)$. We next use $M$ to show that $\mathcal{D}$ satisfies private uniform convergence: let $\mathbb{P}$ be a
761 distribution over $\mathcal{X} \times \{0, 1\}$ and $\epsilon, \delta$ be the error and confidence parameters. Consider the following
762 algorithm:

763 - Draw a sample $S$ from $\mathbb{P}$ of size $m(\epsilon, \delta) = \max\{m_1(\frac{\epsilon}{2}, \frac{\delta}{2}), m_2(\frac{\epsilon}{2}, \frac{\delta}{2})\}$, where

$$m_2 = O\Big(\frac{\mathrm{VC}(\mathcal{D}) + \log(1/\delta)}{\epsilon^2}\Big)$$

764 is the uniform convergence rate of $\mathcal{D}$ (note that by PAC learnability, $\mathrm{VC}(\mathcal{D}) < \infty$).

765 - Apply $M$ on $S$ to obtain an estimator $\mathrm{EST}' : \mathcal{D}' \to [0, 1]$ and output the estimator
766 $\mathrm{EST} : \mathcal{D} \to [0, 1]$ defined by $\mathrm{EST}(d) = \mathrm{EST}'(d')$.

767 We want to show that

$$(\forall d \in \mathcal{D}) : |\mathrm{EST}(d) - L_{\mathbb{P}}(d)| \le \epsilon,$$

768 with probability $1 - \delta$. Indeed, since $m \ge m_2(\frac{\epsilon}{2}, \frac{\delta}{2})$ it follows that

$$(\forall d \in \mathcal{D}) : |L_S(d) - L_{\mathbb{P}}(d)| \le \frac{\epsilon}{2},$$

769 with probability at least $1 - \frac{\delta}{2}$, and since $m \ge m_1(\frac{\epsilon}{2}, \frac{\delta}{2})$,

$$(\forall d \in \mathcal{D}) : |\mathrm{EST}(d) - L_S(d)| = |\mathrm{EST}'(d') - p_S(d')| \qquad \text{(by Eq. (11))}$$
$$\le \epsilon/2,$$

770 with probability $1 - \frac{\delta}{2}$. The desired bound thus follows by a union bound and the triangle inequality.

771 $\qquad\qquad\qquad\qquad\qquad\qquad\qquad\qquad\qquad\qquad\qquad\qquad\qquad\qquad\qquad\qquad\qquad\qquad\qquad\qquad\square$

## C Proof of Corollary 4

773 We begin by defining the predictors $\hat{f}_t$'s that $L$ uses: let $L_0$ be the learner implied by Theorem 3.
774 We first turn $L_0$ into a deterministic learner whose input is $(p_1, y_1), \dots, (p_T, y_T) \in \Delta(\mathcal{W}) \times \{0, 1\}$
775 and that outputs at each iteration $f_t : \mathcal{W} \to [0, 1]$. Then, we extend $f_t$ linearly to $\hat{f}_t$ as discussed in
776 Appendix A.1.1. Let $(p_1, y_1), \dots, (p_T, y_T) \in \Delta(\mathcal{W}) \times \{0, 1\}$, given $w \in \mathcal{W}$, the value $f_t(w)$ is the
777 expected output of the following random process:

778 - sample $w_i \sim p_i$ for $i \le t - 1$,

779 - apply $L_0$ on the sequence $(w_1, y_1), \dots, (w_{t-1}, y_{t-1})$ to obtain the predictor $\tilde{f}_t$, and

780 - output $\tilde{f}_t(x)$.

781 That is,

$$f_t(x) = \mathop{\mathbb{E}}_{w_{1:t-1}}\Big[\mathop{\mathbb{E}}_{\tilde{f}_t \sim L_0}[\tilde{f}_t(w) \mid x_1 \dots x_{t-1}]\Big],$$

782 where $\mathbb{E}_{p_{1:t}}[\cdot]$ denotes the expectation over sampling each $w_i$ from $p_i$ independently, and $\mathbb{E}_{\tilde{f}_t \sim L_0}[\cdot]$
783 denotes the expectation over the internal randomness of the algorithm $L_0$ at iteration $t$. Fi-
784 nally, $\hat{f}_t(p) = \mathbb{E}_{w \sim p}[f_t(w)]$ is the predictor that $L$ uses at the $t$'th round. Note that indeed $\hat{f}_t$
785 is determined (deterministically) from $(p_1, y_1), \dots (p_{t-1}, y_{t-1})$.

We next bound the regret: for every $h \in \mathcal{H}$:

$$\sum_{t=1}^{T} |\hat{f}_t(p_t) - y_t| - |\hat{h}(p_t) - y_t| = \sum_{t:y_t=0} \hat{f}_t(p_t) - \hat{h}(p_t) + \sum_{t:y_t=1} \hat{h}(p_t) - \hat{f}_t(p_t)$$

$$= \sum_{\{t:y_t=0\}} \mathop{\mathbb{E}}_{p_{1:t-1}} \left[ \mathop{\mathbb{E}}_{L_0}[\mathop{\mathbb{E}}_{p_t}[f_t(w_t)] \mid \{w_i\}_{i=1}^{t-1}] \right] - \mathop{\mathbb{E}}_{p_{1:T}} [h(x_t)]$$

$$+ \sum_{\{t:y_t=1\}} \mathop{\mathbb{E}}_{p_{1:T}} [h(w_t)] - \mathop{\mathbb{E}}_{p_{1:t-1}} \left[ \mathop{\mathbb{E}}_{L_0}[\mathop{\mathbb{E}}_{p_t}[f_t(w_t)] \mid \{x_i\}_{i=1}^{t-1}] \right]$$

$$= \sum_{\{t:y_t=0\}} \mathop{\mathbb{E}}_{p_{1:T}} \left[ \mathop{\mathbb{E}}_{L_0}[f_t(x_t) \mid \{w_i\}_{i=1}^{T}] \right] - \mathop{\mathbb{E}}_{p_{1:T}} [h(w_t)]$$

$$+ \sum_{\{t:y_t=1\}} \mathop{\mathbb{E}}_{p_{1:T}} [h(w_t)] - \mathop{\mathbb{E}}_{p_{1:T}} \left[ \mathop{\mathbb{E}}_{L_0}[f_t(w_t) \mid \{w_i\}_{i=1}^{T}] \right]$$

$$= \mathop{\mathbb{E}}_{p_{1:T}} \left[ \mathop{\mathbb{E}}_{L_0}\left[ \sum_{y_t=0} f_t(w_t) - h(x_t) + \sum_{y_t=1} h(w_t) - f_t(w_t) \mid \{w_i\}_{i=1}^{T} \right] \right]$$

$$= \mathop{\mathbb{E}}_{p_{1:T}} \left[ \mathop{\mathbb{E}}_{L_0}\left[ \sum_{t=1}^{T} |f_t(w_t) - y_t| - |h(w_t) - y_t| \mid \{w_i\}_{i=1}^{T} \right] \right]$$

$$\leq \mathop{\mathbb{E}}_{p_{1:T}} \left[ \text{REGRET}_T(L_0, \{w_t, y_t\}_{t=1}^{T}] \right]$$

$$\leq \sqrt{\frac{1}{2}\ell T \log T}.$$

## D  Extending Theorem 2, Item 1 to infinite classes

Here we extend the proof of the upper bound in Theorem 2 to the general case where either $\mathcal{X}$ or $\mathcal{D}$ may be infinite. The proof follows roughly the same lines like the finite case. The first technical milestone we need to consider is to properly define a $\sigma$-algebra over the domain $\mathcal{D}$ and specify the space $\Delta(D)$ of probability measures. For this, we consider $\{0,1\}^{\mathcal{X}}$ as a topological space with an appropriately defined topology and $\Delta(D)$ as the space of Borel-probability measures. We refer the reader to Appendix D.1 for the exact details.

We will also make some technical modifications in the protocol depicted in Fig. 1. The modification is depicted in Fig. 4. The first modification we make is that in the **Else** step, the generator chooses $\bar{d}_t$

---

Consider Fig. 1 with the following modification, at the **Else** Step:

- Find $\bar{d}_t \in \Delta(\mathcal{D})$, **with finite support** such that

$$\left( \forall x \in \mathcal{X} \right) : \mathop{\mathbb{E}}_{d \sim \bar{d}_t} [f_t(d) - x(d)] > \frac{\epsilon}{4}$$

  (if no such $\bar{d}_t$ exists then output *"error"*).

---

Figure 4: Modifying Fig. 1

with finite support. For the finite case, the requirement that $\bar{d}_t$ has finite support is met automatically. The second modification we make allows further slack in the distinguisher. Instead of requiring $> \frac{\epsilon}{2}$ we allow $> \frac{\epsilon}{4}$. Clearly this change in constant does not change the asymptotic regret bound.

**Proof outline.**  To extend the proof to the infinite case it suffices to ensure that the generator in Fig. 1 (with the modification in Fig. 4) never outputs *"error"* in the 2nd item of the "For" loop. To be precise, let us add the following notation that is consistent with the algorithm in Fig. 1. Let $f : \mathcal{D} \to [0,1]$ be measurable.

1. If there exists $p \in \Delta(\mathcal{X})$ such that

$$(\forall d \in \mathcal{D}) : \mathop{\mathbb{E}}_{x \sim p}[f(d) - x(d)] \leq \frac{\epsilon}{2},$$

we say that $f$ satisfies Item 1.

2. If there exists $\bar{d} \in \Delta(\mathcal{D})$ such that

$$(\forall x \in \mathcal{X}) : \mathop{\mathbb{E}}_{d \sim \bar{d}}[f(d) - x(d)] > \frac{\epsilon}{2}$$

we say that $f$ satisfies Item 2.

3. $f$ is *amenable* if it satisfies either Item 1 or Item 2.

When $\mathcal{X}$ and $\mathcal{D}$ are finite, every $f$ satisfies one of Items 1 or 2 (and hence amenable). This is the content of Lemma 5 which is proved using strong duality (in the form of the Minmax Theorem). However, the case when $\mathcal{X}$ and $\mathcal{D}$ are infinite is more subtle. Specifically, the Minmax Theorem does not necessarily hold in this generality.

The next lemma guarantees the existence of a learner $\mathcal{A}$ which only outputs amenable functions. Recall that $\hat{f} : \Delta(\mathcal{D}) \to [0,1]$ denotes the linear extension of $f$ and is defined by $\hat{f}(\bar{d}) = \mathbb{E}_{d \sim \bar{d}}[f(d)]$.

**Lemma 7.** *Let $\mathcal{D}$ be a discriminating class with dual Littlestone dimension $\ell^*$, and let $T$ be the horizon. Then, there exists a deterministic online learning algorithm $\mathcal{A}$ for the dual class $\mathcal{X}$ that receives labelled examples from the domain $\Delta(\mathcal{D})$ and uses predictors of the form $\hat{f}_t$ for some $f_t : \mathcal{D} \to [0,1]$, such that:*

1. *$\mathcal{A}$'s regret is $O(\sqrt{\ell^* T \log T})$, and*

2. *For all $t \leq T$, if the sequence of observed examples $(\bar{d}_1, y_1), \ldots, (\bar{d}_{t-1}, y_{t-1})$ up to iteration $t$, all have finite support then $\mathcal{A}$ chooses $f_t$ that is amenable (in particular $f_1$ is also amenable).*

Our next Lemma shows that Fig. 1 with the modification depicted in Fig. 4 will indeed never output error:

**Lemma 8.** *Consider Fig. 1 with the modification depicted in Fig. 4. Assume $\mathcal{A}$ satisfies the properties in Lemma 7. The for all $t \leq T$ the generator never outputs error.*

*Proof.* The proof follows by induction, for $t = 1$ the amenability of $f_1$ ensures that if $f_1$ doesn't satisfy Item 1 then there exists $\bar{d} \in \Delta(\mathcal{D})$ that satisfy Item 2. Now recall that $\mathcal{X}$ has finite Littlestone dimension and in particular finite VC dimension, by uniform convergence it follow that there is a finite sample $d_1, \ldots, d_m$ such that

$$\sup_{x \in \mathcal{X}} \left| \mathop{\mathbb{E}}_{d \sim \bar{d}}[f_1(d) - x(d)] - \frac{1}{m} \sum_{i=1}^{m} f_1(d_i) - x(d_i) \right| \leq \frac{\epsilon}{4}$$

We then choose $\bar{d}_1$ to be a uniform distribution over $d_1, \ldots, d_m$. By the condition in Item 2 and the above equation we obtain that

$$\mathop{\mathbb{E}}_{d \sim \bar{d}_1}[f(d) - x(d)] > \frac{\epsilon}{4}$$

We continue with the induction step, and consider $t = t_0$. Note that by construction at each iteration up to iteration $t_0$ the algorithm $\mathcal{A}$ observed only distributions with finite support. In particular, we have that $f_{t_0}$ will be amenable. Hence, if it doesn't satisfy Item 1 then we again obtain $\bar{d}$ that satisfies Item 2. We next discretize $\bar{d}$ as before. Using the finite VC dimension of $\mathcal{X}$ we obtain $\bar{d}_{t_0}$ that has finite support and satisfies:

$$\mathop{\mathbb{E}}_{d \sim \bar{d}_{t_0}}[f(d) - x(d)] > \frac{\epsilon}{4}$$

$\square$

Lemma 7, together with Lemma 8, implies the upper bound in Theorem 2, Item 1 via the same argument as in the finite case. This follows by picking the online learner used by the generator in Fig. 1 as in Lemma 7; the amenability of the $f_t$'s (and Lemma 8) implies that the protocol never outputs "error", and the rest of the argument is exactly the same like in the finite case (with slight deterioration in the constants).

**Corollary 5.** *Let $A$ be an algorithm like in the above Lemma. Then, if one uses $A$ as the online learner in the algorithm in Fig. 1, together with the modification in Fig. 4, then the round complexity of it is at most $O(\frac{\ell^*}{\epsilon^2} \log \frac{\ell^*}{\epsilon})$, as in Theorem 2, Item 1.*

In the remainder of this section we prove Lemma 7.

## D.1 Preliminaries

We first present standard notions and facts from topology and functional analysis that will be used. We refer the reader to [35, 34] for further reading.

**Weak\* topology.** Given a compact Haussdorf space $K$, let $\Delta(K)$ denote the space of Borel measures over $K$, and let $C(K)$ denote the space of continuous real functions over $K$. The weak\* topology over $\Delta(K)$ is defined as the weakest[7] topology so that for any continuous function $f \in C(K)$ the following "$\Delta(K) \to \mathbb{R}$" mapping is continuous

$$T_f(\mu) = \int f(k) d\mu(k).$$

We will rely on the following fact, which is a corollary of Banach–Alaglou Theorem (see e.g. Theorem 3.15 in [34]) and the duality between $C(K)$ and $\mathcal{B}(K)$, the class of Borel measures over $K$:

**Claim 2.** *Let $K$ be a compact Haussdorf space. Then $\Delta(K)$ is compact in the weak\* topology.*

**Upper and lower semicontinuity.** Recall that a real function $f$ is called upper semicontinuous (u.s.c) if for every $\alpha \in \mathbb{R}$ the set $\{x : f(x) \geq \alpha\}$ is closed. Note that $\limsup_{x \to x_0} f(x) \leq f(x_0)$ for any $x_0$ in the domain of $f$. Similarly, $f$ is called lower semicontinuous (l.s.c) if $-f$ is u.s.c. We will use the following fact:

**Claim 3.** *Let $K$ be a compact Haussdorf space and assume $E \subseteq K$ is a closed set. Consider the "$\Delta(K) \to [0,1]$" mapping $T_E(\mu) = \mu(E)$. Then $T_E$ is u.s.c with respect to the weak\* topology on $\Delta(X)$.*

*Proof.* This fact can be seen as a corollary of Urysohn's Lemma (Lemma 2.12 in [35]). Indeed, Borel measures are *regular* (see definition 2.15 in [35]. Thus, for every closed set $E$ we have

$$\mu(E) = \inf_{\{U : E \subseteq U, \text{ U is open}\}} \mu(U).$$

Fix a closed set $E$. Urysohn's Lemma implies that for every open set $U \supseteq E$, there exists a continuous function $f_U \in C(K)$ such that $\chi_E \leq f_U \leq \chi_U$, where $\chi_A$ is the indicator function over the set $A$ (i.e. $\chi_A(x) = 1$ if and only if $x \in A$).

Thus, we can write $\mu(E) = \inf_{\{U : E \subseteq U, \text{ U is open}\}} \mu(f_U)$, where $\mu(f_U) = \mathbb{E}_{x \sim \mu}[f_U]$. Now, by continuity of $f_U$, it follows that the mapping $\mu \mapsto \mu(f_U)$ is continuous with respect to the weak\* topology on $\Delta(X)$. Finally, the claim follows since the infimum of continuous functions is u.s.c. $\square$

**Sion's Theorem.** We next state the following generalization of Von-Neumann's Theorem for u.s.c/l.s.c payoff functions.

**Theorem 4** (Sion's Theorem). *Let $W$ be a compact convex subset of a linear topological space and $U$ a convex subset of a linear topological space. If $F$ is a real valued function on $W \times U$ with*

- *$F(w, \cdot)$ is l.s.c and convex on $U$ and*

- *$F(\cdot, u)$ is u.s.c and concave on $W$*

*then,*

$$\max_{w \in W} \inf_{u \in U} F(w, u) = \inf_{u \in U} \max_{w \in W} F(w, u)$$

**Tychonof's space.**  The last notion we introduce is the topology we will use on $\{0,1\}^{\mathcal{X}}$. Given an arbitrary set $\mathcal{X}$, the space $\mathcal{F} = \{0,1\}^{\mathcal{X}}$ is the space of all functions $f : X \to \{0,1\}$. The product topology on $\mathcal{F}$ is the weakest topology such that for every $x \in \mathcal{X}$ the mapping $\Pi_x : \mathcal{F} \to \{0,1\}$, defined by $\Pi_x(f) = f(x)$ is continuous.

A basis of open sets in the product topology is provided by the sets $U_{x_1,\ldots,x_m}(g)$ of the form:
$$U_{x_1,\ldots,x_m}(g) = \{f : g(x_i) = f(x_i)\ i = 1,\ldots,m\},$$
where $x_1,\ldots,x_m$ are arbitrary elements in $X$ and $g \in \mathcal{F}$.

A remarkable fact about the product topology is that the space $\mathcal{F}$ is compact for any domain $\mathcal{X}$ (see for example [27]). We summarize the above discussion in the following claim

**Claim 4.** *Let $\mathcal{X}$ be an arbitrary set and consider $\mathcal{F} = \{0,1\}^{\mathcal{X}}$ equipped with the product topology. Then $\mathcal{F}$ is compact and $\Pi_x \in C(\mathcal{F})$ for every $x \in X$, where $\Pi_x$ is defined as $\Pi_x(f) = f(x)$.*

## D.2  Two Technical Lemmas

The proof of Lemma 7 follows from the following two Lemmas. Throughout the proofs we will treat $\mathcal{D}$ as a topological subpace in $\{0,1\}^{\mathcal{X}}$ with the product topology. We will also naturally treat $\Delta(\mathcal{D})$ as a topological space equipped with the weak* topology.

**Lemma 9** (Analog of Lemma 5). *Assume $\mathcal{D} \subseteq \{0,1\}^{\mathcal{X}}$ is closed and let $f : \mathcal{D} \to [0,1]$. Assume that $\hat{f}$ is u.s.c (with respect to the weak* topology on $\Delta(\mathcal{D})$) then $f$ is amenable.*

**Lemma 10** (Analog of Corollary 4). *Let $\mathcal{D} \subseteq \{0,1\}^{\mathcal{X}}$ be closed and let $\ell^*$ denote its dual Littlestone dimension. Then, there exists a deterministic online learner that receives labelled examples from the domain $\Delta(\mathcal{D})$ such that for every sequence $(p_t, y_t)_{t=1}^T$ we have that:*
$$\mathrm{REGRET}_T(L) \le \sqrt{\frac{1}{2}\ell T \log T}$$

*Moreover, at each iteration $t$ the predictor, $\hat{f}_t$, used by $L$ is of the form $\hat{f}_t\left[\bar{d}\right] = \mathbb{E}_{d \sim \bar{d}}(f_t(d))$ for some $f_t : \mathcal{D} \to [0,1]$. Finally, for every $t \le T$, if the sequence of observed examples $(\bar{d}_1, y_1),\ldots,(\bar{d}_{t-1}, y_{t-1})$ all have finite support then $\hat{f}_t$ is u.s.c.*

We first show how to conclude the proof of Lemma 7 using these lemmas and later prove the two lemmas.

**Concluding the proof of Lemma 7.**  The proof follows directly from the two preceding Lemmas. Given a discriminating class $\mathcal{D} \subseteq \{0,1\}^{\mathcal{X}}$ there is no loss of generality in assuming $\mathcal{D}$ is closed, since closing the class with respect to the product topology does not increase its dual LIttlestone dimension.

Now, take the learner $\mathcal{A}$ whose existence follows from Lemma 10. Since each $\hat{f}_t$ is u.s.c we obtain via Lemma 9 that each $f_t$ is also amenable.

**Proof of Lemma 9.**  Lemma 9 extends Lemma 5 to the infinite case. Similar to the proof of Lemma 5 which hinges on Von-Neumann's Minmax Theorem, the proof here hinges on Sion's Theorem which is valid in this setting.

Before proceeding with the proof we add the following notation: let $\mathbb{R}^{\mathcal{X}}_{fin}$ denote the space of real-valued functions $v : \mathcal{X} \to \mathbb{R}$ with finite support, i.e. $v(x) = 0$ except for maybe a finite many $x \in \mathcal{X}$. We equip $\mathbb{R}^{\mathcal{X}}_{fin}$ with the topology induced by the $\ell_1$ norm, namely a basis of open sets is given by the open balls $U_{v,\epsilon} = \{u : \sum_{x \in \mathcal{X}} |v(x) - u(x)| < \epsilon\}$. $\mathbb{R}_{fin}(\mathcal{X})$ is indeed a linear topological space (i.e. the vector addition and scalar multiplication mappings are continuous). Finally, define
$$\Delta_{fin}(\mathcal{X}) := \{p \in \mathbb{R}^{\mathcal{X}}_{fin} : p(x) \ge 0 \sum_{x \in \mathcal{X}} p(x) = 1\}.$$

Next, let $f : \mathcal{D} \to [0,1]$ be such that $\hat{f}$ is u.s.c. Our goal is to show that $f$ is amenable. Set $F$ to be the following real-valued function over $\Delta(\mathcal{D}) \times \Delta_{fin}(\mathcal{X})$:
$$F(\bar{d}, p) = \mathbb{E}_{\bar{d} \sim d}\left[f(d) - \sum_{x \in \mathcal{X}} p(x) x(d)\right]$$

It suffices to show that

$$\max_{\bar{d}\in\Delta(\mathcal{D})}\ \inf_{p\in\Delta_{fin}(\mathcal{X})} F(\bar{d},x) = \inf_{p\in\Delta_{fin}(\mathcal{X})}\ \max_{\bar{d}\in\Delta(\mathcal{D})} F(\bar{d},p) \tag{12}$$

Indeed, the assumption that Item 1 does not hold implies in particular that

$$\inf_{p\in\Delta_{fin}(\mathcal{X})}\ \max_{d\in\Delta(\mathcal{D})} F(\bar{d},p) \geq \frac{\epsilon}{2}.$$

Eq. (12) then states that

$$\max_{\bar{d}\in\Delta(\mathcal{D})}\ \inf_{x\in\mathcal{X}}\ \mathbb{E}_{d\sim\bar{d}}\left[f(d)-x(d)\right] \geq \frac{\epsilon}{2}.$$

which proves that Item 2 holds.

Eq. (12) follows by an application of Theorem 4 on the function $F$. Thus, we next show the premise of Theorem 4 is satisfied by $F$. Indeed, $W = \Delta(\mathcal{D})$ is compact and convex, and $U = \Delta_{fin}(\mathcal{X})$ is convex. We show that $F(\cdot,p)$ is concave and u.s.c for every fixed $p \in \Delta_{fin}(\mathcal{X})$: indeed, $F(\cdot,p)$ is in fact linear and therefore concave. We show that $F(\cdot,p)$ is u.s.c by showing that it is the sum of (i) a u.s.c function (i.e. $\mathbb{E}_{d\sim\bar{d}}[f(d)]$) and (ii) finitely many continuous functions (i.e. $\sum_{x\in\mathcal{X}} p(x)\,\mathbb{E}_{d\sim\bar{d}}[x(d)]$). Indeed, (i) by assumption $\hat{f}(\bar{d}) = \mathbb{E}_{d\sim\bar{d}}[f(d)]$ is u.s.c, and (ii) by Claim 4, the mapping $\Pi_x(d)$ is continuous for every $x \in \mathcal{X}$ which, by the definition of the weak* topology, implies that $\bar{d} \to \mathbb{E}_{d\sim\bar{d}}\,\Pi_x(d) = \mathbb{E}_{d\sim\bar{d}}\,[x(d)]$ is continuous.

Finally, because $\mathbb{E}_{d\sim\bar{d}}[x(d)] \leq 1$ is bounded, it follows that $F(\bar{d},\cdot)$ is linear and continuous in $p$ for every fixed $\bar{d}$: indeed treating $\hat{f}(\bar{d})$ and $\{\mathbb{E}_{\bar{d}\sim d}\,[x(d)]\}_{x\in\mathcal{X}}$ as bounded constants, we have that:

$$F(\bar{d},p) = \hat{f}(\bar{d}) - \sum_{x\in X} p(x)\,\mathbb{E}_{\bar{d}\sim d}\,[x(d)]$$

**Proof of Lemma 10.** Lemma 10 follows from a close examination of the proof provided in [6] for Theorem 3 and the extension to Corollary 4.

The fact that the learner outputs a predictor of the form $\hat{f}_t = \mathbb{E}_{\bar{d}\sim d}\,[f_t(d)]$ follows by construction in Corollary 4. So, it suffices to show that the $f_t$'s can be chosen to be u.s.c. Call a function $s : \mathcal{D} \to \{0,1\}$ an SOA-type function if there exists a hypothesis class $\mathcal{H} \subseteq \mathcal{X}$ such that

$$s(d) = \begin{cases} 0 & \mathrm{Ldim}(\mathcal{H}|_{(d,0)}) = \mathrm{Ldim}(H) \\ 1 & \text{else} \end{cases}$$

where $H|_{(d,0)} = \{h \in H\} : h(d) = 0\}$.

In the proof by [6] of Theorem 3 the authors construct an online learner which at each iteration uses a randomized predictor (i.e. a distribution over predictors). One can observe and see that this randomized predictor only uses SOA-type function: namely, the algorithm holds, at each iteration, a distribution $q_t$ over a finite set of SOA type functions $\{s_k\}$, and at each iteration picks the prediction made by $s_k$ with probability $q_t(s_k)$.

The extension in Corollary 4 of this predictor to the domain $\Delta(\mathcal{D})$ is done by choosing:

$$f_t(d) = \mathbb{E}_{\bar{d}_{1:T}}\left[\mathbb{E}_{s\sim L_0}\,[s(d)|d_1,\ldots,d_{t-1}]\right] = \mathbb{E}_{\bar{d}_{1:T}}\left[\sum q_t(s_k)s_k(d)|d_1,\ldots,d_{t-1}\right]$$

Namely, the choice of $f_t$ is the expectation over the algorithm's prediction, taking expectation both over the choice of the algorithm and over the sequence of observations. $d_1,\ldots,d_{t-1}$, drawn according to $\bar{d}_1,\ldots,\bar{d}_{t-1}$. Now because $\bar{d}_1,\ldots\bar{d}_{t-1}$ all have finite support we can summarize these expectations and write:

$$f_t = \sum \lambda_k s_k,$$

for some choice of SOA-type functions and weights $\lambda_k \geq 0$.

Since the sum of u.s.c functions is u.s.c and since the multiplication of a u.s.c function with positive scalar is u.s.c, it is enough to prove that every SOA-type function $s$ induces an u.s.c function over $\Delta(\mathcal{D})$ via the identification $\mu \mapsto \mu\left(\{d : s(d) = 1\}\right)$. By Claim 3 it is enough to show that the set

$s^{-1}(0)$ is open. To this end we show that for every $d \in s^{-1}(0)$ there is an open neighborhood of $d$ which is contained in $s^{-1}(0)$. Indeed, if $d \in s^{-1}(0)$, then there exist $x_1, \ldots, x_{2^\ell}$ that $d(x_i) = 0$ for all $i$, and they shatter a tree. Consider the open neighborhood of $d$ defined by $U = \cap_i \{d : d(x_i) = 0\}$. $U \subseteq s^{-1}(0)$ since if there were $d' \in U$ such that $s(d') = 1$ then $\mathrm{Ldim}(\mathcal{H}|_{(d',0)}) < \mathrm{Ldim}(\mathcal{H}) = \ell$. However, since $d' \in U$ then $x_1, \ldots, x_{2^\ell} \in \mathcal{H}|_{(d',0)}$ and they shatter a tree of depth $\ell$ which is a contradiction.