[Reviews · NeurIPS 2020]

Review 1

Summary and Contributions: This paper provides the first domain-size independent sample complexity bounds for DP-Fooling, and shows an equivalence between private synthetic data generation and private learning. Besides, this paper introduces and characterizes a new class of SDGs and demonstrates their utility in the construction of private synthetic data.

Strengths: As an analogy to the relationship between PAC learning and fooling, this paper shows an equivalence between private synthetic generation and private PAC learning. Meanwhile, it shows an equivalence between sequentially-foolability and a finite Littlestone dimension. I think the problems are important theoretical problems in DP and these discoveries are interesting.

Weaknesses: Although this paper has interesting discoveries, I am not sure of the theoretical contributions of the paper. I think the main theoretical contribution is that it has proved the connection between sequential-foolability and a finite Littlestone dimension, which is kind of intuitive.

Correctness: Yes.

Clarity: Generally speaking, I think the paper is well-written. The flow is good and the paper is easy to understand. Besides, there are few typos in the paper. I think it will be better if a proof sketch can be given in Section 4.

Relation to Prior Work: I think this paper gives a good reference to the previous work. The only citation in my mind which is missing is Vietri, Giuseppe, et al. "New Oracle-Efficient Algorithms for Private Synthetic Data Release." arXiv preprint arXiv:2007.05453 (2020).

Reproducibility: Yes

Additional Feedback:


Review 2

Summary and Contributions: This highly theoretical paper investigates the conceptual relationship between a series of properties of a class of predicates (boolean-valued functions). These properties are: 1) Whether the class of predicates is privately and properly learnable via probably approximately correct learning, 2) Whether differentially private synthetic data generation is possible that produces synthetic data that is on average close to the real data as measured by all predicates of the class, 3) Whether a differentially private estimation algorithm exists that can produce good estimates for the fraction of samples that satisfy each predicate, 4) Whether there is a differentially private algorithm that can produce estimators in the sense of uniform convergence. The main result of the paper is to show that these properties (1 to 4) are equivalent for all classes of predicates .

Strengths: This paper is theoretically very well grounded. It explores the relationship between seemingly different properties of classes of predicates, which generalizes to plenty of tasks we might want to tackle via machine learning. Thus, I think the paper is relevant to the community. To the best of my knowledge, this result is novel.

Weaknesses: I'm not quite sure how relevant the findings are in any practical sense; the "equivalence" shown here can be quite different from equivalent in practice, due to prohibitively large constant factors. For all machine learning tasks I am aware of, even small constants make a significant difference and finding any reasonable utility-privacy trade-off is already difficult. I think the paper could benefit from making these relations more explicit. While subsampling does indeed allow for better privacy bounds, it also requires a much larger dataset to subsample from and at least in practice there are massive hurdles to, say, asking for a 1000 times larger dataset.

Correctness: I didn't find a flaw, but I have not checked all of the proofs thoroughly.

Clarity: I found it difficult to read the paper, which might be because it is an in-depth analysis of complex theoretical relationships. Maybe some emphasis could be made to rewrite the earlier sections, adding some intuition about what exactly the goal here is and introducing all acronyms.

Relation to Prior Work: I am not very familiar with this type of theoretical predicate class property relations, but from what I can see, it seems the authors discussed their contributions clearly.

Reproducibility: Yes

Additional Feedback: The first few pages of the paper are in stark contrast to the high-level discussion at the end. In the discussion you give a very good and intuitive explanation of what differentially private synthetic data can provide. Maybe some of those intuitions (or at least the spirit of writing in such an intuitive manner) could be moved to the earlier sections of the paper, specifically sections 2 and 3. Overall I think this is an interesting theoretical finding and if I understand it correctly, it can inspire further research on the theoretical limitations (and possibilities) of differentially private data synthesis. After author response: I agree that theoretical bounds are important, which is why I think the paper could be accepted even if the authors cannot produce evidence for the practical relevance of their results.


Review 3

Summary and Contributions: This paper studies differentially private synthetic data generation with respect to a (possibly infinite) class of queries D and shows that if D is privately and properly PAC learnable, then it admits a private Synthetic Data Generator (SDG). More generally, the paper establishes an equivalence between the statements: 1. D is privately and properly PAC learnable, 2. D admits a private SGD, and 3. D is sanitizable. The proof introduces, as an intermediate step, sequential SDGs, and shows an equivalence between classes that admit sequential SDGs and classes with finite Littlestone dimension.

Strengths: This paper studies differentially private SDG, which is an area of interest to the NeurIPS community. -It attempts to provide a characterization of the classes that admit a private SDG, via private learnability, finite Littlestone dimension, and the notion of sequential SDGs. The latter is introduced in this paper and seems interesting in its own right. I think its importance stems from the fact that it connects and provides equivalences among many interesting established notions. -Previous work on private SDGs requires that the size of the original dataset depend logarithmically on the size of the class. This work gives sample complexity bounds for private SDGs for *infinite* classes, which depends on the Littlestone dimension of the class. -Although it might seem that in view of recent results, the connection of private SDGs and sequential SDGs/online learnability/Littlestone dimension is natural, the proof is fairly intricate and combines several ideas from these areas.

Weaknesses: -There is no consideration for computational or communication efficiency in the constructions for a private/sequential SDG. The generator and discriminators are assumed to be omnipotent in terms of computational power. So at this point the paper serves more as a proof-of-concept and not as a practical construction.

Correctness: The paper fully proves all of its claims. Disclaimer: I am not too familiar with some topological results so although the proof for the extension to infinite classes (Appendix D) seems reasonable to me, there is a chance I missed something there.

Clarity: In general the paper is well written and includes some intuition about its constructions. There is room for improvement however (please see suggestions below).

Relation to Prior Work: Prior work is discussed and indeed to the best of my knowledge there are no prior sample complexity upper bounds for infinite query classes. However, I would have preferred a separate section for related work, where there would be -a bit more discussion on the (cited) recent works on the connection of Littlestone dimension and private learnability, which seems to be a parallel and closely related connection. -a statement of the best known bounds for finite classes. -the related work on practical constructions of SDGs that are in the Discussion.

Reproducibility: Yes

Additional Feedback: The paper's presentation is very good and the proof sketches/overviews are helpful! Main suggestions: -I would have liked an actual statement of the sample complexity! I know it's mostly a qualitative result, but a quantitative statement does not appear except for deep down in the proofs. A quantitative statement would help identify the gap and motivate future work. -Similarly, I think a discussion on the computational inefficiency of the algorithms and the particular barriers where this arises would help. -The format and more importantly the positioning of the algorithms (Figures 1,2,3 and the fooling algorithm in page 18) is quite inconvenient. An algorithmic environment instead of bullet-points would make them more readable. Also, they seem to be in random pages. Please move them closer to the proofs of the relevant corollaries/theorems. -Restating the theorems before proving them would also help. -Is it necessary to define both sequential foolability and eps-sequential foolability? I found that a bit confusing/unnecessary but maybe there is a reason we need both. Lower-level comments: -Although [41] is en excellent survey to refer the reader to, some basic (e.g. post-processing) results first appear in [14]. -Reference [36] has a typo in one of the authors' name and also it's a FOCS paper so perhaps it's best to cite the conference version instead of the arxiv (other instances of this appear). -Line 56, 242: In the supplementary, some compilation issues appear ("??") when referring the reader to specific sections of the appendix. -Line 77, 450: Prelimineries -> Preliminaries -Line 159: sample complexity should have the VC(D)/eps^2 term -Line 217: SDG -that- with -Line 218: exists->exist -Line 466: denoted -> denote -Line 476: W^M->W^m -Line 511: in advanced -> in advance -Line 664: Inorder -> In order -Line 781: Shouldn't it be: Expectation over p_{1:t-1}, f_t(x) conditioned on w_1, ..., w_{t-1}? I think there's a few more instances of mixing x and w in the proof of Corollary 4. A suggestion for related work: A recent work that you *needn't* have cited as it was just published at ICML2020, but is related (on private data release with some connections to Littlestone dimension) is the one by Bassily, Cheu, Moran, Nikolov, Ullman, Wu. Also, not as closely related but still on private SDGs, Vietri, Tian, Bun, Steinke, Wu, at the same conference. =========================================== Thank you for your response. I support acceptance of this paper and hope you can indeed incorporate these changes in the presentation (perhaps more importantly to include a discussion of the sample complexity - and the gap).


Review 4

Summary and Contributions: - Private-proper-agnostic learnability implies private-fool-ability (and hence is equivlant to it and to sanitize-ability). - Definition of sequential-SDG. - Round complexity upper and lower bounds for sequential-SDG in term of dual-Littlestone dimension. The bounds gives characterization to classes which obtain a sequential-SDG as Littlestone lasses.

Strengths: The paper is well written. The definition of sequential-S looks like an interesting online-variant to the known SDG problem. The results are solid and adds a nice level on top of the recent results in the field.

Weaknesses: Most of the equivalences are known (the authors themselves note that). Edit: the authors addressed this point in their feedback and as I said also pointed out to the novel part and the equivalences not previously known. My score stays the same.

Correctness: Seems so

Clarity: Yes

Relation to Prior Work: Yes

Reproducibility: Yes

Additional Feedback:


Review 5

Summary and Contributions: The paper provides a theoretical framework to study how many IID samples from some target distribution are needed to generate differentially private (DP) synthetic data that is indistinguishable from the actual data w.r.t a fixed class of statistical queries. Since prior solutions consider finite query classes, they generalize previous techniques to work for infinite query classes. Then the main question they address is what classes of statistical queries can be privately approximated with a polynomial number of IID samples from the actual distribution, or by their terminology, what classes can be privately fooled (DP-foolable classes). They show that the classes that can be privately fooled are the same classes that are privately and proper PAC learnable. They extend their results by providing class equivalences to other learning tasks. The sample complexity bounds in previous work (for finite classes) depend on the size of the query class. Here they show that if the query class is privately (proper) PAC learnable, then the sample complexity is independent of the cardinally of the query class. Instead, their bounds depend on the Littlestone dimension of the query class. The approach in prior work on synthetic data generation consists of a sequential game between a data generator player and a discriminator. On each round, the generator proposes a distribution to approximate the real distribution. Then the generator finds a query (from a fixed query class) with a high discrepancy between the real and fake data. A query class is sequentially foolable if the game converges to finding a distribution that is close to the true target distribution. The main result from theorem 1 is: A class that is private (proper) PAC learnable implies finite Littlestone dimension (this follows from Alon et al. 2018); therefore, it is sequentially foolable. Then a class that is sequentially foolable and private (proper) PAC learnable can be used to construct the converging sequential game, as described in the above paragraph. Hence a private (proper) PAC learnable is DP-foolable.

Strengths: The paper contributes a helpful theoretical framework unifying private PAC learning with the theory behind DP synthetic data generation. They provide the first sample complexity that does not depend on the size of the query class.

Weaknesses: The framework in this paper seems to apply only to binary classes, but we might also be interested in multi-label and continuous classes. The lower bound of theorem 2 seems very weak. The upper bound applies to all epsilon, but the lower bound to epsilon less than 1/2, and the lower bound seems not to have a dependence on epsilon. Perhaps more details and explanations should be included in the main paper to understand the problem's difficulty. Also, authors may want to include some discussion about closing the sample complexity gap.

Correctness: Yes. However, I may have missed details in the supplementary material.

Clarity: Yes. However, the authors deferred most technical details to the supplement. The main paper should provide some more intuition or proof sketches of their primary technical results.

Relation to Prior Work: This paper generalizes an established methodology for DP synthetic data generation to include infinite classes.

Reproducibility: Yes

Additional Feedback: The paper would benefit from including some examples of DP-foolable classes that are common in the ML literature. What is the implication to private PAC learning under framework? How many samples do we need to generate DP synthetic data that can be used to train a model to epsilon accuracy?

[Author Response · NeurIPS 2020]

We thank the reviewers for their positive comments. In the text below we address specific concerns
raised by the reviewers – we apologize for conciseness; we will address all issues raised by the
reviewers, including fixing typos, adding citations.

**Reviewer 1**

• **I think the main theoretical contribution is that it has proved the connection between**
**sequential-foolability and a finite Littlestone dimension, which is kind of intuitive.**
The main contribution is the equivalence between DP PAC learning, DP Fooling, and Sequential-
Fooling. Perhaps the main technical milestone in the derivation of this equivalence is to prove
that Littlestone dimension implies sequential foolability (without dependence on the domain-size).
While this implication may be intuitive, its proof does not follow immediately from existing results
(that exhibited domain-size dependent bounds). We discuss this between lines 36-57. (While we
promised a more detailed discussion on the technical difficulties in the full version, due to some
technical error this was neglected from the submitted supplementary – this will be corrected and a
more detailed discussion will be added).

**Reviewer 2**

• **the "equivalence" shown here can be quite different from equivalent in practice, due to pro-**
**hibitively large constant factors**
True, but a first step in understanding the sample complexity of many practical tasks is to provide
some non-trivial bounds, and understand existing theoretical limitations.

**Reviewer 3**

• **The generator and discriminators are assumed to be omnipotent in terms of computational**
**power.**
Yes, in this work we study only the sample complexity in disregard of computational issues. Clearly,
computational issues change the picture, nevertheless analysing the sample complexity in this
simplistic setting is an important first step.

•

**1. a bit more discussion on the (cited) recent works on the connection of Littlestone dimen-**
**sion...**
**2. a statement of the best known bounds for finite classes.**
**3. the related work on practical constructions of SDGs that are in the Discussion...**
These are good comments, and we will add these important details. We will also follow the rest
of the suggestions made by the reviewer- explicit sample complexity, discussion on computational
pitfalls, take care of formatting issues and restatements, as well as relate to the citations suggested.

**Reviewer 4**

• **Most of the equivalences are known (the authors themselves note that).**
Many of the derivations we discuss are due to previous work (and we appropriately cite). Neverthe-
less this paper contains several new contributions. Particularly, that Littlestone dimension implies
sequential foolability was not derived in previous work (without dependence on the domain-size)..

**Reviewer 5**

• **The framework in this paper seems to apply only to binary classes**
This is true. Extending the results to other domains does seem like an interesting direction for future
research.
• **The lower bound of theorem 2 seems very weak. The upper bound applies to all epsilon, but**
**the lower bound to epsilon less than 1/2**
$\epsilon \leq 1/2$ is indeed tight and we can (will) provide an example where $\epsilon \geq 1/2$ is achieved trivially.
Regarding the gap between the sample complexities, this is indeed open and we will discuss this-
thanks for pointing that out.



[Meta-Review · NeurIPS 2020]

The reviewers agree that this paper makes a solid contribution to the literature of differentially private synthetic data. In particular, the paper establishes new connection with private PAC learning for new regimes. The authors should follow reviewers' suggestions to revise the paper and consider citing suggested related work.